# Variability of cholesterol accessibility in human red blood cells measured using a bacterial cholesterol-binding toxin

**Rima S Chakrabarti[1†], Sally A Ingham[1†], Julia Kozlitina[2], Austin Gay[2], Jonathan C Cohen[2\*], Arun Radhakrishnan[3\*], Helen H Hobbs[1,3\*]**

[1]Howard Hughes Medical Institute, University of Texas Southwestern Medical Center, Dallas, United States; [2]Departments of Internal Medicine, University of Texas Southwestern Medical Center, Dallas, United States; [3]Molecular Genetics, University of Texas Southwestern Medical Center, Dallas, United States

**Abstract** Cholesterol partitions into accessible and sequestered pools in cell membranes. Here, we describe a new assay using fluorescently-tagged anthrolysin O, a cholesterol-binding bacterial toxin, to measure accessible cholesterol in human red blood cells (RBCs). Accessible cholesterol levels were stable within individuals, but varied >10 fold among individuals. Significant variation was observed among ethnic groups (Blacks>Hispanics>Whites). Variation in accessibility of RBC cholesterol was unrelated to the cholesterol content of RBCs or plasma, but was associated with the phospholipid composition of the RBC membranes and with plasma triglyceride levels. Pronase treatment of RBCs only modestly altered cholesterol accessibility. Individuals on hemodialysis, who have an unexplained increase in atherosclerotic risk, had significantly higher RBC cholesterol accessibility. Our data indicate that RBC accessible cholesterol is a stable phenotype with significant inter-individual variability. Factors both intrinsic and extrinsic to the RBC contribute to variation in its accessibility. This assay provides a new tool to assess cholesterol homeostasis among tissues in humans.

**\*For correspondence:** jonathan.
cohen@utsouthwestern.edu (JCC);
arun.radhakrishnan@
utsouthwestern.edu (AR); helen.
hobbs@utsouthwestern.edu (HHH)

[†]These authors contributed
equally to this work

**Competing interest:** See
page 24

**Reviewing editor:** Stephen G
Young, University of California,
Los Angeles, United States

## Introduction

Cholesterol is an essential component of vertebrate cell membranes (*Goldstein et al., 1979*). Membrane cholesterol is supplied by the diet or synthesized endogenously from acetyl coenzyme A. All cells in the body except red blood cells (RBCs) synthesize cholesterol. To maintain cholesterol balance, an amount of cholesterol equivalent to the sum of what is made in the cells and acquired from the diet must be removed from the body. The major pathway for cholesterol excretion in humans is via the biliary system. The proteins and processes by which cholesterol is secreted into the bile, either directly, or after conversion to bile acids, have been well characterized (*Berge et al., 2000*; *Russell, 2009*). In contrast, the process by which cholesterol from peripheral tissues is delivered to the liver for excretion remains poorly defined.

A mechanism for the centripetal trafficking of cholesterol from extrahepatic tissues to the liver, which has been termed reverse cholesterol transport, was first conceptualized by *Glomset (1968)*. That work, and subsequent studies, have implicated high density lipoproteins (HDL) as an essential component of this pathway (*Bailey, 1964*) (*Figure 1*). In the first step of the Glomset model, HDL acquires free cholesterol from cell membranes in peripheral tissues. Free cholesterol is transferred from cells to HDL by ATP binding cassette, subfamily A, member 1 (ABCA1) (*Lawn et al., 1999*). The free cholesterol in the HDL particle is then esterified with long-chain fatty acids by lecithin-cholesterol acyltransferase (LCAT) (*Glomset, 1968*) and sequestered in a hydrophobic core below the

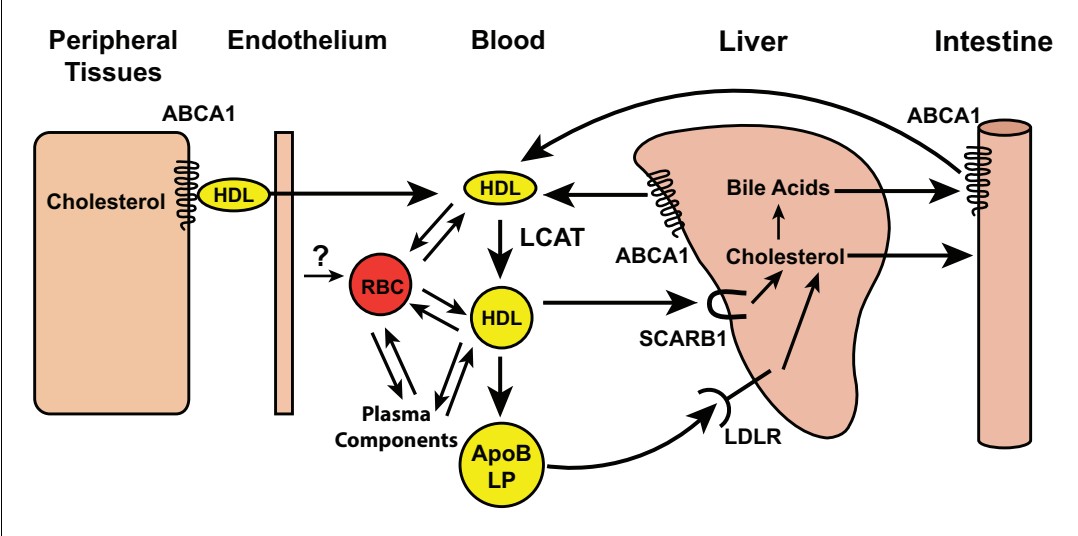

**Figure 1.** Schematic of reverse cholesterol transport in the blood. Free cholesterol in peripheral tissues is effluxed by ABCA1 transporters to high-density lipoprotein (HDL), and transported into the vascular space. Lecithin cholesterol acyl transferase (LCAT), present on HDL, esterifies free cholesterol. The cholesterol esters formed by the LCAT reaction are either taken up by SCARB1 receptors on hepatocytes or transferred to apolipoprotein B containing lipoproteins (ApoB LP) and taken up into hepatocytes by the low density lipoprotein receptor (LDLR). Red blood cells (RBCs) may also participate in reverse cholesterol transport by accepting cholesterol from vascular endothelial cells and participating in a cholesterol exchange with lipoproteins and plasma components, such as albumin.

phospholipid surface of HDL (*Stoffel et al., 1974*). This step is critical to the delivery of cholesterol to the liver; sequestration in the HDL core decreases the propensity of cholesterol to exchange with cell membranes, thus limiting the retrograde transfer of cholesterol from HDL back to the tissues. The cholesteryl ester is then delivered to the liver either directly via a cell surface receptor, scavenger receptor class B member 1 (SCARB1) (*Acton et al., 1996*), or indirectly after transferring the cholesterol to Apolipoprotein (Apo)B-containing lipoproteins, which then undergo endocytosis by the low density lipoprotein receptor (LDLR) (*Figure 1*). Several components of the HDL-mediated reverse cholesterol transport pathway have been supported by studies in cultured cells and in vivo (*Lewis and Rader, 2005*).

Despite data implicating HDL as a central component of the reverse cholesterol transport pathway, some observations in murine models of HDL deficiency are not compatible with an essential role for HDL in the transport of cholesterol from peripheral tissues to the liver and bile (*Xie et al., 2009*; *Jolley et al., 1998*) (*Figure 1*). Mice lacking ABCA1, or ApoA1, the major structural protein of HDL, have very low HDL levels and yet have no alterations in cholesterol turnover in peripheral tissues (*Xie et al., 2009*; *Jolley et al., 1998*).

Glomset proposed that RBCs may participate in the transport of cholesterol from tissues to the liver (*Glomset, 1968*). Approximately 50% of circulating cholesterol is carried in RBC membranes and tracer studies in humans indicate that the magnitude of the cholesterol flux through RBCs is comparable to the total efflux of free cholesterol from tissues (*Turner et al., 2012*). Cholesterol exchanges freely between RBCs, plasma, circulating lipoproteins, and cells (*Hagerman and Gould, 1951*) (*Figure 1*). More recently, Smith and colleagues (*Hung et al., 2012*) showed that reducing the number of RBCs in ApoA1 knockout mice, which have little to no circulating HDL, decreases the transfer of radiolabeled cholesterol from macrophages to feces.

For RBCs to participate in reverse cholesterol transport, the cholesterol must be located in the outer leaflet of the RBC membrane, and thus accessible for transfer to proteins and cells. The accessibility of cholesterol in a membrane is related to its chemical activity (*McConnell and Radhakrishnan, 2003*), alternatively referred to as its fugacity (tendency to flee). The chemical activity of cholesterol generally increases with increasing cholesterol concentration. However, at lower concentrations, complex formation with phospholipids can suppress the chemical activity of cholesterol

below what would be expected in the absence of lipid interactions. At higher concentrations, when phospholipids become limiting, the chemical activity of cholesterol rises. Sharp rises in chemical activity have been detected in model membranes by measurements of cholesterol accessibility, as assayed by cyclodextrin extraction (*Radhakrishnan and McConnell, 2000*; *Litz et al., 2016*), oxidation by cholesterol oxidase (*Ahn and Sampson, 2004*; *Lange et al., 2013*), and binding of cholesterol-dependent cytolysins such as perfringolysin O (PFO) or anthrolysin O (ALO) (*Heuck et al., 2000*; *Gay et al., 2015*). A sharp rise in the chemical activity of RBC membrane cholesterol has been observed using the cholesterol oxidase assay to assess cholesterol accessibility (*Lange et al., 1980*). In all of these cases, the increase in cholesterol accessibility with increasing cholesterol concentration was sigmoidal, rather than linear, with little change until a threshold concentration was reached. Once the cholesterol content of the membrane exceeded this threshold, accessibility of membrane cholesterol increased sharply.

The partitioning of cholesterol between accessible and sequestered pools may allow RBCs to serve as vehicles for cholesterol transport from peripheral tissues to the liver. As a first step towards elucidating the relative contributions of the accessible and sequestered pools of RBC cholesterol to cholesterol transport, we have developed an assay using ALO, a cholesterol-binding toxin produced by *Bacillus anthracis* (*Bourdeau et al., 2009*), to measure the relative amount of accessible cholesterol in RBC membranes.

Using this assay, we show that RBC cholesterol accessibility is stable within individuals, but varies over a more than ten-fold range between individuals. To address the basis for the inter-individual variation in accessible RBC cholesterol, we examined the effect of demographic, physiologic, and biochemical differences among individuals on this newly described trait. Whereas age and gender had modest effects on accessible RBC cholesterol, significant ethnic differences were apparent in RBC cholesterol accessibility. The variation in cholesterol accessibility was not related to the cholesterol content of the membrane or to pronase-accessible proteins in the RBC membrane, but rather to the content of other membrane lipids and to levels of circulating triglycerides (TG). Finally, we find that a group of patients on hemodialysis, who have more coronary heart disease than anticipated based on known risk factors (*Levey et al., 1998*), have higher RBC cholesterol accessibility.

## Results

### Measurement of RBC cholesterol accessibility using cholesterol binding domain of ALO

As described previously, a recombinant protein consisting of the cholesterol binding domain 4 (D4) of ALO was purified from *E. coli* and covalently modified with a maleimide-linked fluorescent dye (Alexa Fluor 488) (*Gay et al., 2015*). This reagent, which does not lyse RBCs (*Gay et al., 2015*), is referred to hereafter as fALOD4 (*Figure 2A*). Aliquots of $2.5 \times 10^5$ RBCs were incubated with increasing amounts of fALOD4 for 3 hr prior to analysis by flow cytometry. The forward and side light scatter patterns and the median fluorescence per RBC are shown in *Figure 2B*. Dose-response and time course experiments were performed to determine the initial conditions for the assay (*Figure 2C*). The extent of fALOD4 binding per cell increased with increasing concentrations of fALOD4 (left panel). Even at the highest concentrations of fALOD4 tested, binding did not saturate. A value within the linear range of fALOD4 binding (250 nM) was used to perform the time-course experiment (right panel). At this concentration, binding approached saturation after 3 hr, and this incubation time was chosen for subsequent assays. These assay conditions did not result in significant hemolysis (<5%) (*Figure 2C*).

### RBC cholesterol accessibility was not linearly related to RBC cholesterol content

Next we assessed the effect of the cholesterol content of the RBCs on fALOD4 binding. RBCs were incubated with either hydroxypropyl-$\beta$-cyclodextrin (HPCD) or cholesterol/methyl-$\beta$-cyclodextrin (MCD) complexes to reduce or increase the cholesterol content of RBC membranes, respectively (*Figure 2D*, left panel). No fALOD4 binding was observed when the cholesterol content of the RBCs was reduced below ~46 mole%. Above this apparent threshold, the binding increased sharply. When the data were plotted on a log-linear scale the relationship between cholesterol content of the

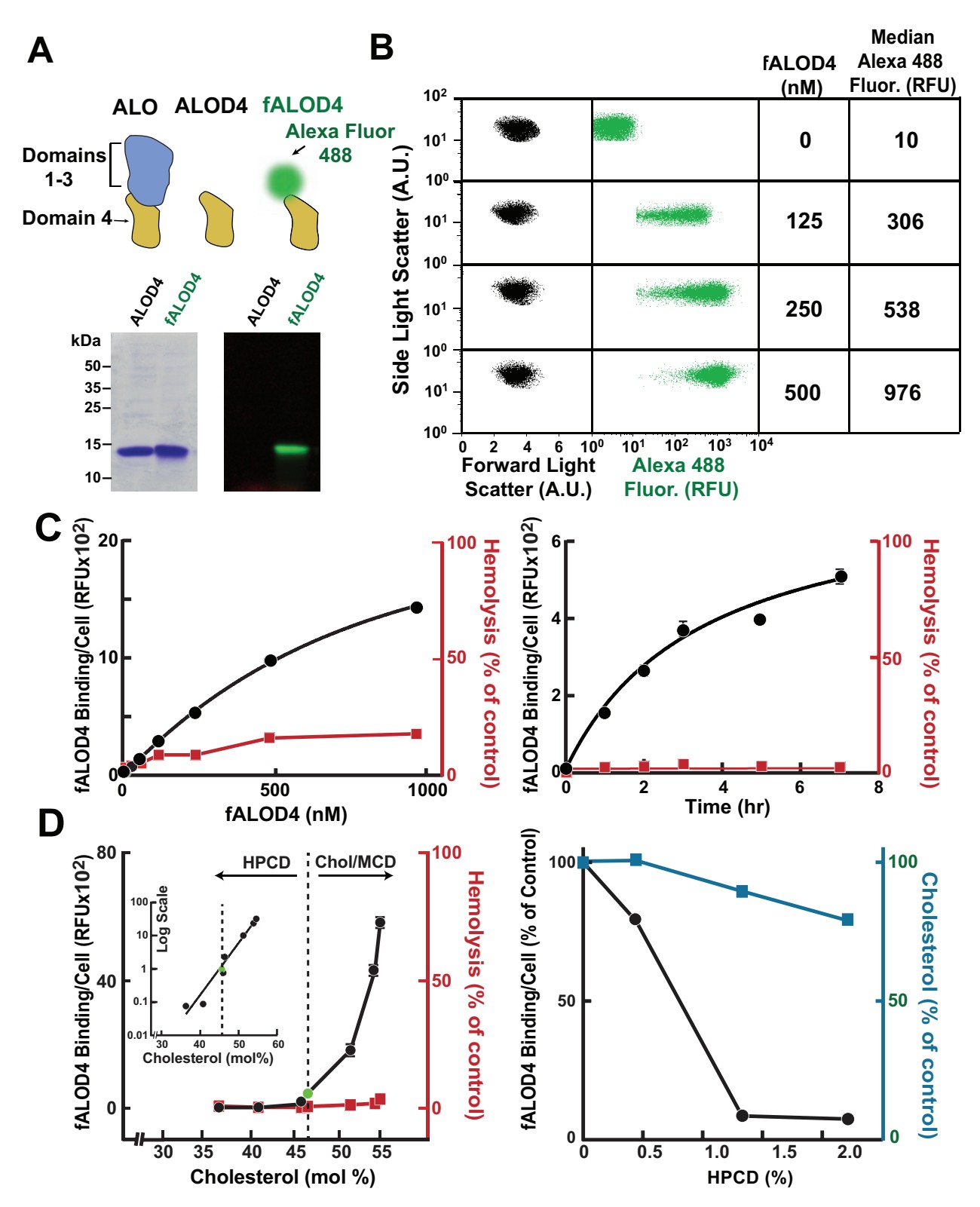

**Figure 2.** Assay for red blood cell (RBC) cholesterol accessibility. (**A**) Schematic of anthrolysin O (ALO) domains. Domain 4 (ALOD4) binds cholesterol but does not oligomerize or form membrane-lysing pores. In fALOD4, Alexa Fluor 488 dye is covalently attached to an engineered cysteine near the NH₂-terminus of domain 4. Purified proteins (5 μg) were subjected to SDS-PAGE (15%) and visualized by Coomassie staining (*left*) or fluorescence scanning (LI-COR) at 600 nm (*right*). (**B**) Flow cytometry analysis of fALOD4 binding to RBCs. RBCs (2.5 × 10⁵ RBCs in 500 μl buffer D) were incubated

*Figure 2 continued on next page*

*Figure 2 continued*

for 3 hr at 4°C with fALOD4 at the indicated concentrations. Fluorescence was measured by a FACSCalibur flow cytometer as described in the Materials and methods. Forward light scatter (FSC), side light scatter (SSC), and Alexa 488 fluorescence measurements for 10,000 RBCs were acquired on the flow cytometer. Median Alexa 488 fluorescence per cell was calculated using FlowJo software. AU, arbitrary units; RFU, relative fluorescence units. (C) Dose response (*left*) and time course (*right*) of fALOD4 binding to RBCs. Each reaction was set up as described above using either the indicated concentrations of fALOD4 (*right*) or 250 nM fALOD4 (*left*). After incubation at 4°C for 3 hr (*left*) or for the indicated times (*right*), fALOD4 RBC binding was measured by flow cytometry. Hemolysis during fALOD4 binding reactions was determined by measuring the release of hemoglobin as described in the Materials and methods. 100% hemolysis is defined as the amount of hemoglobin released after treatment of RBCs with 1% (w/v) Triton X-100. Data points represent means of three independent measurements of the same sample (technical replicates). Error bars, which are often not visible due to the small variation, represent the SEM. The experiment was repeated three times and the results were similar. (D) Effect of RBC cholesterol modulation on RBC fALOD4 binding. (*Left*) RBCs were not treated (green circle) or treated with either hydroxypropyl-β-cyclodextrin (HPCD) or cholesterol/methyl-β-cyclodextrin (MCD) to reduce or increase the cholesterol content of RBCs. The fALOD4 binding assay was performed as described in the Materials and methods using 250 nM of fALOD4. Lipids were extracted from ghost membranes isolated from the RBCs, and the molar percentage of cholesterol was measured as described in the Materials and methods. The dashed line indicates the cholesterol content of untreated RBCs. Hemolysis was measured as described in the Materials and methods. Inset: Data for the experiment plotted on a logarithmic-linear scale. (*Right*) RBCs were treated with HPCD to remove cholesterol and both fALOD4 binding and cholesterol content were measured as described in the Materials and methods. The fALOD4 binding per cell and cholesterol content (mole %) in untreated RBCs was set to 100%. Data points represent the mean of three independent measurements of fALOD4 binding and a single measurement of RBC cholesterol content. Error bars represent the SEM. RFU, relative fluorescence units. The experiments were repeated three times and the results were similar.

membranes and fALOD4 binding was linear (*Figure 2D*, inset, left panel). In cells incubated with increasing concentrations of HPCD, a small reduction in cholesterol content of the cell was associated with a marked reduction in fALOD4 binding (*Figure 2D*, right panel). Such sigmoidal dependences on cholesterol concentration have been observed previously for the binding of cholesterol-binding toxins like ALO and PFO to model liposomes (thresholds vary from 20–50 mole% depending on the phospholipid content) (*Gay et al., 2015*; *Nelson et al., 2008*), to plasma membranes (PM) of human fibroblasts (threshold ~35 mole%) (*Das et al., 2013*), and to purified ER membranes from cultured CHO-K1 cells (threshold ~5 mole%) (*Sokolov and Radhakrishnan, 2010*). Thus, the characteristics of the assay conform to expectations based on prior experiments using membranes from several different cell types and cellular organelles.

## RBC cholesterol accessibility varied between individuals in a reproducible and stable manner

We then measured fALOD4 binding to RBCs isolated from unrelated healthy individuals. Duplicate measurements were made on each sample. The data were expressed relative to a control sample that was used in all assays and the results were expressed as normalized relative fluorescence units (nRFU). fALOD4 binding varied over a 10-fold range (from 0.2 to 2.6) among samples from different individuals but the correlation between the two measurements on the same sample was very high (R = 0.94) (*Figure 3A*). The intra-assay coefficient of variation for the assay was 10.7% ± 5.3%.

To determine if fALOD4 binding is stable over time in individuals, blood was drawn from five people on three separate days over a 25-day interval (*Figure 3B*). The intra-individual variation in fALOD4 binding per cell was 8.8% ± 4.2% (range: 3.6–12.8%). Thus, the inter-individual variation in fALOD4 binding per RBC was much greater than the variability seen in any given individual.

Representative data showing the distributions of fALOD4 binding to RBCs for an individual with low (top), medium (middle), and high (bottom) levels of fALOD4 RBC binding are shown in *Figure 4A*. Although the distribution of cells by size is similar in the three samples (left), the distributions of fluorescence levels progressively shifted to the right. Since the distribution of signal is skewed, the median rather than the mean values of fALOD4 binding are used in the studies to follow.

We next tested whether variations in fALOD4 binding arose because the fALOD4 probe was present in limiting amounts. For this experiment, we used RBCs from three individuals whose cholesterol accessibility measurements fell in the low, medium, and high ranges (*Figure 4B*). We incubated increasing amounts of RBCs with a fixed concentration of fALOD4 (250 nM). At RBC concentrations of less than $10^6$ RBCs/assay, fALOD4 binding values clearly separated into the low, medium, and high ranges. At higher RBC concentrations, fALOD4 binding per cell decreased in all cases and

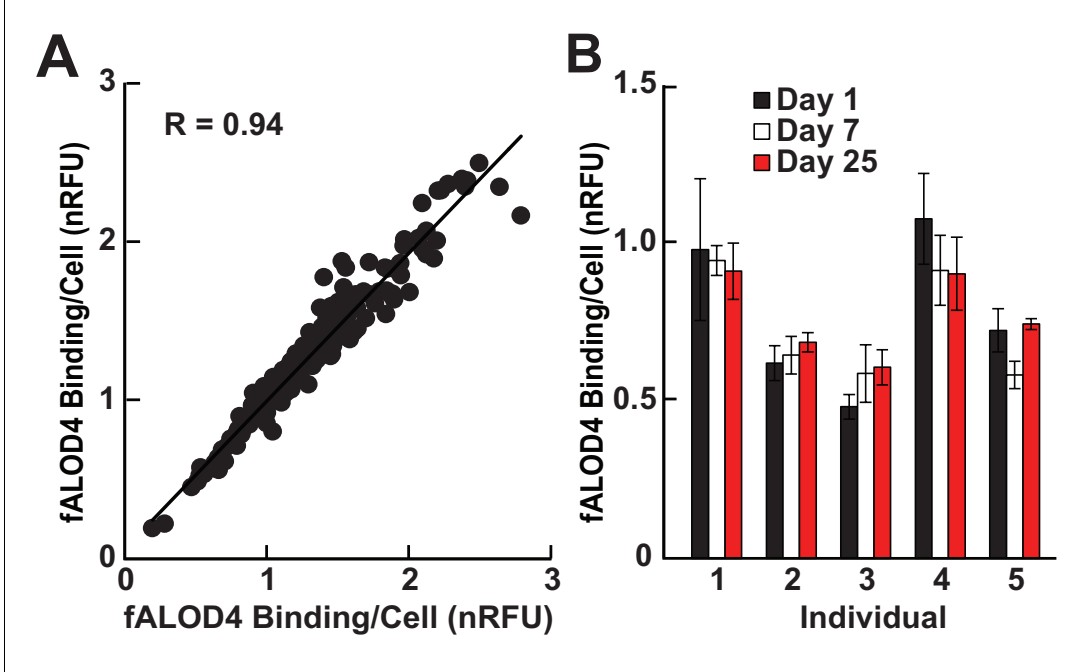

**Figure 3.** Characterization of RBC cholesterol accessibility assay. (**A**) Intra-assay correlation was determined by measuring fALOD4 binding to two aliquots of RBCs from 164 unrelated healthy individuals as described in the Materials and methods. The levels of fALOD4 binding were normalized to a reference sample of RBCs from a healthy individual that was included in all the assays. Data points represent the mean of three independent measurements from each of the RBC aliquots. (**B**) Intra-individual variability of fALOD4 binding to RBCs from five individuals. Blood samples from five unrelated individuals were collected on three separate days over a month and fALOD4 binding was measured. fALOD4 binding values in **A** and **B** were normalized to the binding values obtained from the reference blood sample. The reference blood sample was collected from the same healthy volunteer prior to each experiment in this study and the sample was processed and assayed concurrently. Error bars represent the mean ± SEM of three measurements from each blood sample.

resolution among high, medium and low binding was lost. Therefore, In all subsequent assays, $2.5 \times 10^5$ RBCs/assay were used to ensure that fALOD4 was not limiting. In this experiment, the levels of fALOD4 binding were not normalized and were expressed in relative fluorescent units (RFUs).

## RBC cholesterol accessibility was not related to the cholesterol content of the membranes

To examine the relationship between fALOD4 binding and the cholesterol content of the RBC membranes, we prepared duplicate aliquots of RBCs from 73 individuals. One aliquot was used to determine fALOD4 binding. The second aliquot was used to prepare ghost membranes, extract lipids, and measure cholesterol concentration. A control sample was included in each assay and was used to normalize measurements. Surprisingly, no relationship was found between the cholesterol content of the RBC membranes and normalized fALOD4 binding (*Figure 4C*). Thus, fALOD4 binding does not simply reflect the cholesterol content of the RBC membrane.

## Significant ethnic differences in RBC cholesterol accessibility

Next we examined the effect of major demographic variables on RBC cholesterol accessibility. We assayed RBCs from 364 men and women of self-assigned ancestry: Whites (European-American, n = 182), Hispanics (n = 84) and Blacks (African-American, n = 98). Blood samples were obtained either at a local blood bank or as part of a sample of healthy volunteers who were enrolling in the Dallas BioBank. A summary of the demographic and clinical variables in the sample is provided in *Table 1*. The ethnic breakdown was 50% White, 27% Black and 23% Hispanic. The Hispanics in the samples were significantly younger than the other two groups and there was a significantly lower proportion of men in the Black and Hispanic groups. The hematocrits (Hct), hemoglobin (Hb), RBC

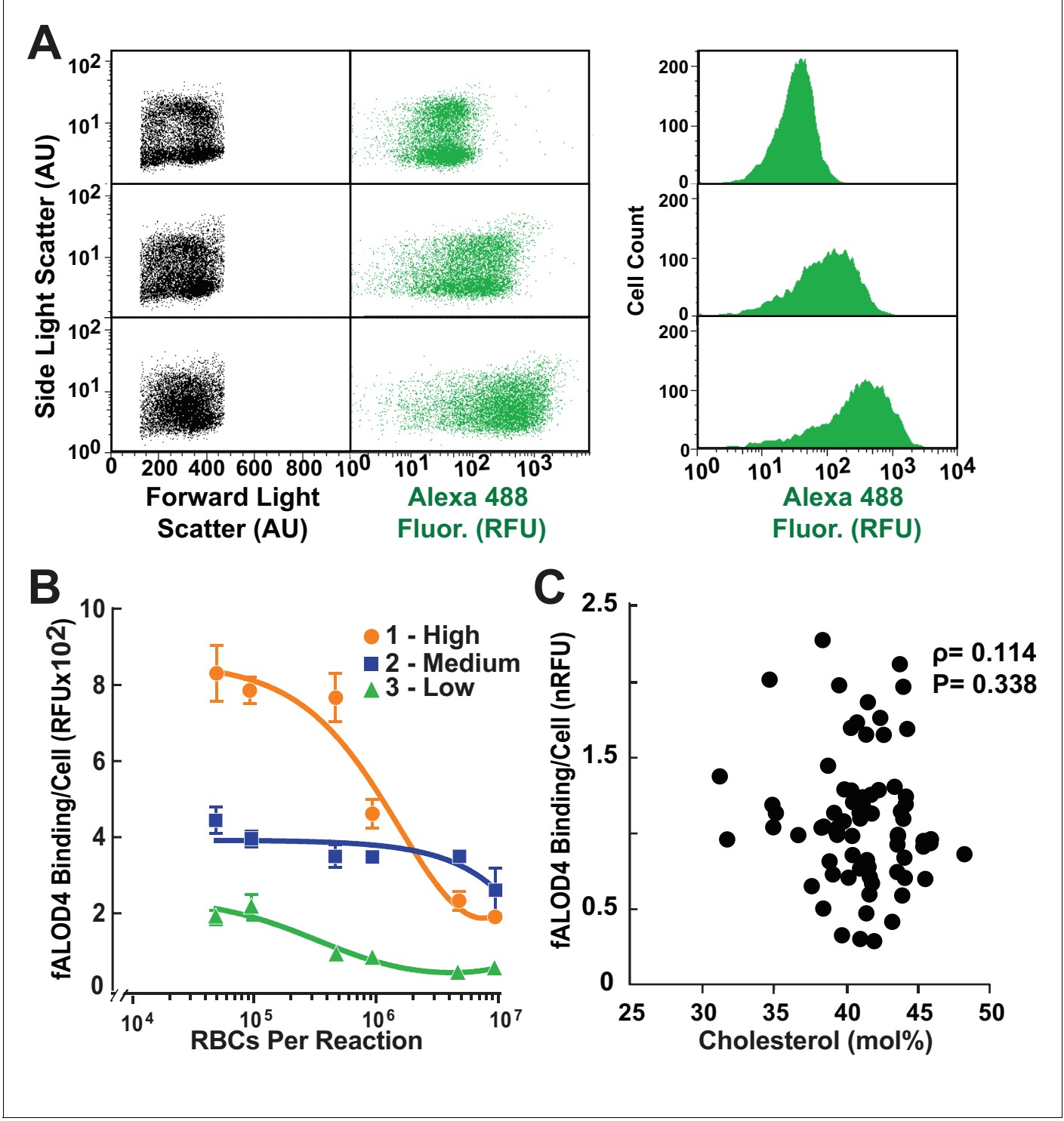

**Figure 4.** Distribution of fALOD4 binding to RBCs in individuals with low, medium and high binding. (**A**) Flow cytometry analysis of fALOD4 binding to RBCs from three individuals with low (top), medium (middle) and high (bottom) fALOD4 binding. Forward light scatter, side light scatter, and Alexa 488 fluorescence measurements of 10,000 RBCs were acquired using a FACSCalibur flow cytometer as described in the Materials and methods. Fluorescence data are presented as both a dot plot (middle) and histogram (right). AU, arbitrary units; RFU, relative fluorescence units. (**B**) Relationship between RBC number and fALOD4 binding in three individuals with low (green), medium (blue) and high (orange) binding. Samples were collected on the same day and the fALOD4 binding assay was performed as described in the legend to *Figure 1*. Error bars represent the mean ± SEM of three measurements from each blood sample. The experiment was repeated once and the results were similar. (**C**) RBC cholesterol content and fALOD4 binding to RBCs. RBC ghost membranes were prepared from RBCs of 73 healthy, unrelated individuals. Total lipids were extracted from the

*Figure 4 continued on next page*

*Figure 4 continued*
membranes and the molar percentage of cholesterol was measured as described in the Materials and methods. fALOD4 binding values were normalized to the binding values obtained from the reference blood sample. Shown is the Spearman correlation between fALOD4 binding, normalized to the reference blood sample, and RBC membrane total cholesterol expressed as mole % of total lipids. Data points represent the means of three independent measurements of fALOD4 binding and a single measurement of RBC cholesterol content. The experiment was repeated once and the results were similar. nRFU, normalized relative fluorescence units.

counts, and plasma cholesterol levels were similar among the three groups. Whites had a significantly lower mean corpuscular Hb concentration (MCHC) and mean plasma levels of HDL-C and TG differed significantly among the three groups.

To determine which of the inter-related factors listed in *Table 1* independently contributed to differences in fALOD4 binding, we performed multivariate analysis. Accessible RBC cholesterol levels did not vary significantly with age (*Table 2*). Men had significantly higher levels of accessible RBC cholesterol (p=0.01), although these differences were modest compared to the variation in levels among the three ethnic groups (*Figure 5* and *Table 2*). Since relative binding differed so significantly among the three ethnic groups (Blacks vs. Whites, p=2.83–13; Hispanics vs. Whites, p=1.6E-02), we plotted the data for each group separately. In all three groups, the distribution of fALOD4 binding to RBCs was positively skewed and varied over a >10 fold range (*Figure 5*). Median fALOD4 binding was lowest in Whites (0.77 nRFU), intermediate in Hispanics (0.86 nRFU), and highest in Blacks (1.18 nRFU) (p<0.0001).

No significant association was found between fALOD4 binding and RBC indices, including hematocrit, Hb concentration, RBC count, MCHC, mean corpuscular volume (MCV) or RBC distribution width (RDW), after adjustment for age, gender, and ethnicity (*Table 1* and *Figure 6*). Thus, the differences in RBC cholesterol accessibility levels among individuals or among the three different ethnicities did not appear to be related to differences in the gross morphology or Hb content of the RBC.

**Table 1.** Characteristics of 364 healthy subjects. Data are mean ± SD or median (25th – 75th percentile), unless otherwise indicated. Blood samples were obtained from healthy subjects after an 8 hr fast. Plasma lipids and lipoproteins and RBC indices were measured by Quest Diagnostics as described in the Materials and methods. Continuous variables were compared between groups using ANOVA adjusted for age and gender, where appropriate and categorical variables were compared using chi-square tests. Variables with non-normal distributions were inverse-normally transformed before analysis. Hct, hematocrit; Hb, hemoglobin; MCHC, mean corpuscular hemoglobin concentration; MCV, mean corpuscular volume; RDW, RBC distribution width; HDL-C, high density lipoprotein cholesterol; LDL-C, low density lipoprotein cholesterol.

| Characteristic | n | All (n = 364) | White (n = 182) | Black (n = 98) | Hispanic (n = 84) | P-value |
|---|---|---|---|---|---|---|
| Age, years | 364 | 40.9 ± 16.5 | 42.2 ± 17.7 | 44.9 ± 12 | 33.4 ± 16.2 | 3.5E-06 |
| Male, n (%) | 364 | 136 (37.4) | 91 (50) | 19 (19.4) | 26 (31) | 1.1E-06 |
| Female, n (%) | 364 | 228 (62.6) | 91 (50) | 79 (80.6) | 58 (69) | 1.1E-06 |
| Hct (%) | 345 | 41.1 ± 5.6 | 41.4 ± 6.1 | 40.6 ± 5.5 | 40.8 ± 4.5 | 0.27 |
| Hb (g/dL) | 344 | 13.2 ± 1.9 | 13.5 ± 1.9 | 12.9 ± 1.8 | 13 ± 2.0 | 0.98 |
| RBC count ($10^6/\mu$L) | 345 | 4.6 ± 0.6 | 4.7 ± 0.6 | 4.6 ± 0.6 | 4.6 ± 0.5 | 0.065 |
| MCHC | 344 | 32.3 ± 0.8 | 31.8 ± 0.7 | 32.6 ± 0.8 | 32.2 ± 0.8 | 7.5E-10 |
| MCV (fL) | 345 | 87.9 ± 8.3 | 88.3 ± 8.8 | 87.2 ± 9.5 | 88.0 ± 5.3 | 0.23 |
| RDW | 345 | 14.4 ± 1.8 | 14.7 ± 1.7 | 14.2 ± 1.9 | 14.3 ± 1.7 | 0.11 |
| Total cholesterol (mg/dL) | 363 | 178.4 ± 36.8 | 180.3 ± 37.3 | 179.6 ± 34.6 | 173.1 ± 38.0 | 0.47 |
| HDL-C (mg/dL) | 345 | 51.9 ± 14.4 | 50.3 ± 14.6 | 57 ± 14 | 49.1 ± 13.2 | 0.012 |
| LDL-C (mg/dL) | 341 | 101.4 ± 35.4 | 99.5 ± 32.2 | 102.9 ± 31.6 | 103.3 ± 44.6 | 0.53 |
| Triglycerides (mg/dL) | 345 | 76 (105-164) | 129 (85-184) | 83 (67-117) | 106 (85-159) | 2.8E-07 |

**Table 2.** Factors associated with fALOD4 binding in 364 healthy subjects. Beta coefficients are in SD units. For quantitative variables (age and lipid levels), betas are given per 1 SD change in the predictor. Partial R$^2$ indicates the proportion of variance in fALOD4 binding explained by a given predictor after adjusting for other factors.

| Factor | Beta | SE | P-value | Partial R$^2$ (%) |
|---|---|---|---|---|
| Age | 0.017 | 0.023 | 0.46 | 0.16 |
| Male Gender | 0.119 | 0.046 | 0.010 | 1.93 |
| Ethnicity: | | | | |
| White | Reference | – | – | |
| Black | 0.403 | 0.053 | 2.8E-13 | 14.63 |
| Hispanic | 0.130 | 0.053 | 1.6E-02 | |
| Triglycerides (mg/dL) | −0.089 | 0.024 | 0.00021 | 3.99 |
| Total cholesterol (mg/dL) | 0.053 | 0.024 | 0.026 | 1.45 |
| Total | | | | 21.61 |

## fALOD4 binding was inversely related to plasma TG levels

We also examined the relationship between fALOD4 binding and plasma lipid and lipoprotein levels after adjusting for age, gender and ethnicity. fALOD4 binding to RBCs was not related significantly to plasma levels of total cholesterol, low-density lipoprotein-cholesterol (LDL-C) or high-density lipoprotein (HDL)-C levels but was inversely related (p=0.0018) to plasma TG levels (*Figure 7*).

Ethnicity accounted for the largest fraction of variation in fALOD4 binding. The proportion of variance in fALOD4 binding that was explained by ethnicity in a model adjusted for age, sex, and plasma lipid levels was 14.63% (*Table 2*). Only plasma TG (p=0.00021) and total cholesterol levels (p=0.026) contributed significantly to the variance of fALOD4 binding in a multivariable model adjusted for demographic factors (*Table 2*). Together, age, gender, race, plasma TG and cholesterol levels accounted for 21.61% of the inter-individual variability in fALOD4 binding. None of the other factors listed in *Table 1* (RBC indices, Hb, Hct, plasma levels of LDL-C or HDL-C) were significantly associated with fALOD4 binding after adjusting for the above covariates.

Next we performed studies to determine the effects of differences in factors intrinsic to the RBC membrane on fALOD4 binding.

## Relationship between fALOD4 binding and RBC membrane proteins

To determine if RBC membrane proteins contribute to the differences in RBC cholesterol accessibility, we labeled the membrane proteins with biotin and then treated the biotin-labeled RBCs with pronase (*Figure 8A*). Western blotting was performed using streptavidin-HRP to assess the effect of pronase treatment on accessible membrane proteins. The protein content of the RBC membranes decreased progressively with increasing amounts of pronase. Hemolysis measurements show that the integrity of the RBC membranes was not significantly affected by the pronase treatment, except at the highest level tested. At the lowest concentration of pronase (0.02 µg/mL), fALOD4 binding increased by ~25% (*Figure 8A*). Higher concentrations of pronase did not further increase fALOD4 binding despite further decreases in membrane protein content. Based on these results we concluded that the protein composition of the RBC membrane is not a major determinant of fALOD4 binding, although we cannot exclude the possibility that a protein(s) that is resistant to pronase treatment contributes to the inter-individual differences in fALOD4 binding.

We also examined the relationship between fALOD4 binding and ABO blood groups in participants that were recruited at a blood donation facility (*Figure 8B* and *Table 3*). All of these study subjects were White. No significant differences were apparent between individuals with blood types A, B, and O, whereas individuals who were Rhesus antigen negative (Rh-) had significantly lower fALOD4 binding than did individuals who were Rh+. The relationship between blood type and accessible cholesterol binding will need to be confirmed in an independent sample and examined in other ethnic groups.

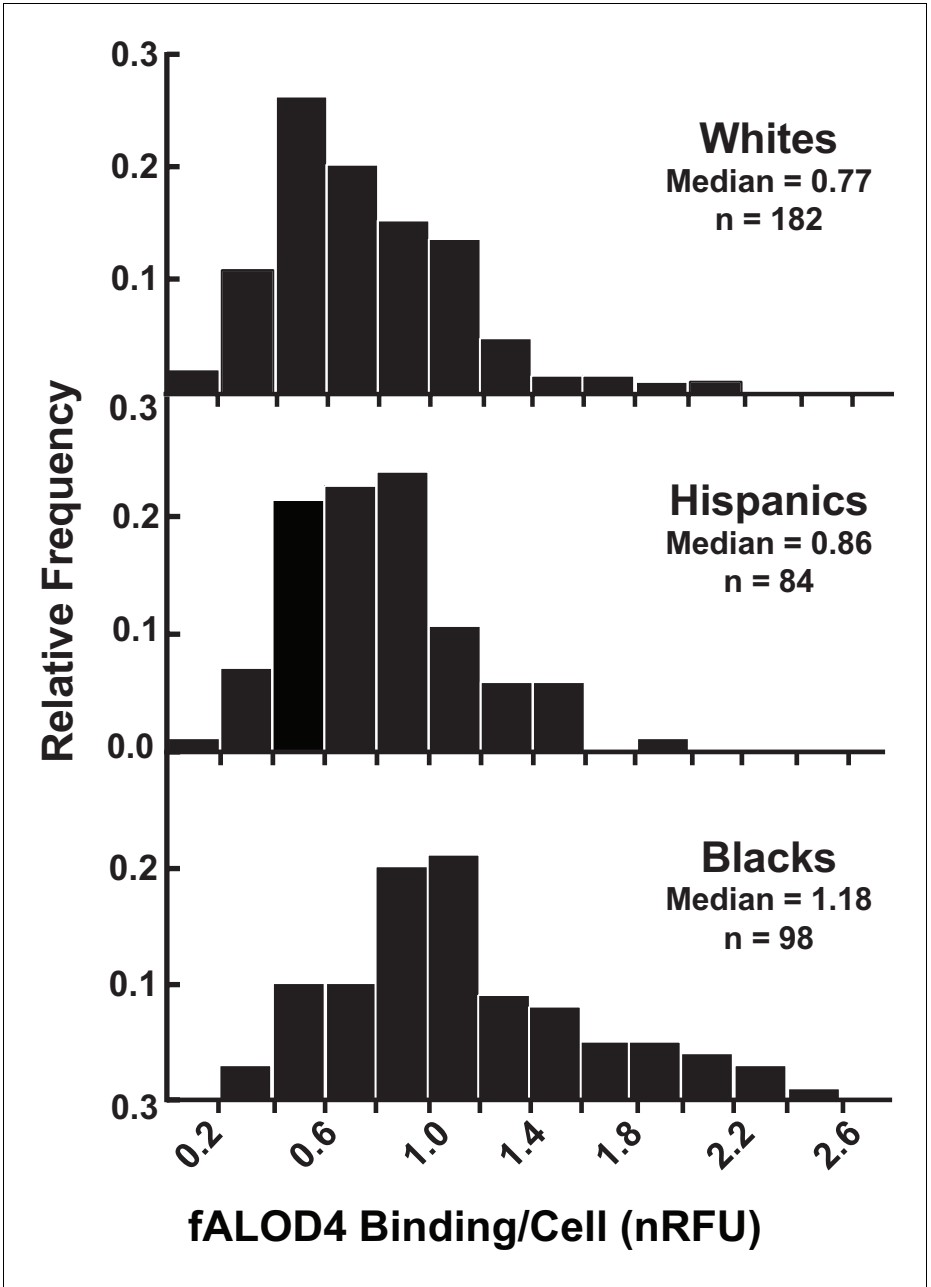

**Figure 5.** Distribution of fALOD4 binding to RBCs from 364 healthy, unrelated individuals. Blood samples were collected from 364 individuals and RBCs were isolated as described in the Materials and methods. Each measurement was performed in triplicate. fALOD4 binding values were normalized to those of the reference blood sample. Data are plotted as frequency histograms of median fALOD4 binding. nRFU, normalized relative fluorescence units.

## fALOD4 binding increases with treatment of RBCs with phospholipase A2 (PLA2) and sphingomyelinase (SMase)

To determine if the phospholipid composition of the RBC membrane contributes to fALOD4 binding, we treated RBCs with PLA2, which hydrolyzes the ester linkage at the sn-2 position of phospholipids, or with SMase, which hydrolyzes the ester linkage between the sn-1 carbon and phosphate group of sphingomyelin (SM). fALOD4 binding increased progressively when the RBCs were incubated with increasing amounts of PLA2 (*Figure 9A*). Increased fALOD4 binding was also seen after

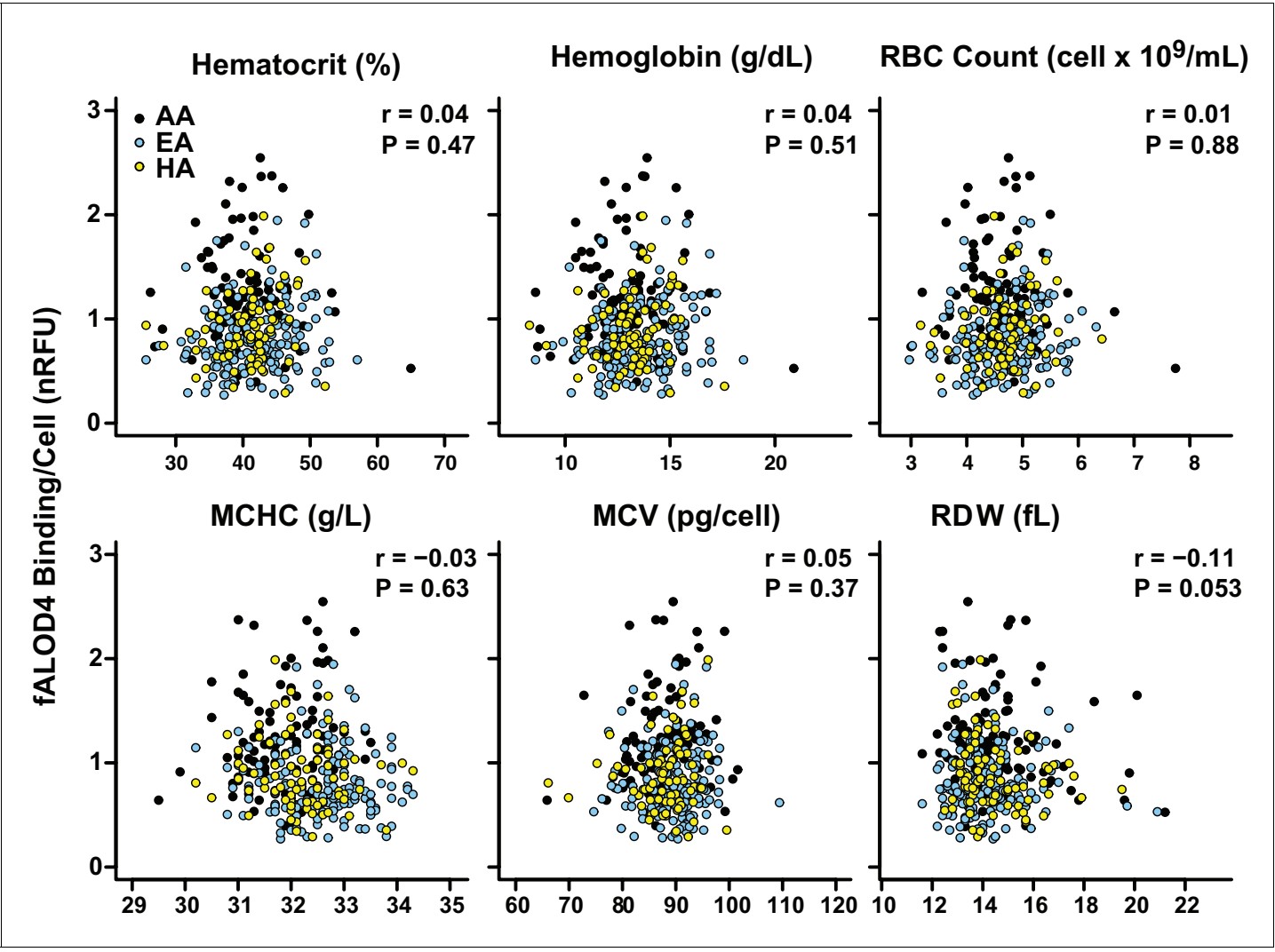

**Figure 6.** Correlation between fALOD4 binding to RBCs and RBC indices. Hematocrit, hemoglobin, RBC count, mean corpuscular hemoglobin concentration (MCHC), mean corpuscular volume (MCV), and RBC distribution width (RDW) were measured by Quest Diagnostics. fALOD4 binding to RBCs was measured in triplicate as described in the Materials and methods and normalized to the reference blood sample. Spearman's correlation coefficients (r) and p-values were calculated using the software package Prism 6 (two tailed T-test).

the cells were incubated with SMase (*Figure 9B*), which is consistent with previous observations that SMase treatment increased cholesterol accessibility to PFO in fibroblasts (*Das et al., 2013*), increased MCD extraction of cholesterol from erythrocytes (*Besenicar et al., 1778*), and increased PM to ER trafficking of cholesterol in fibroblasts (*Slotte and Bierman, 1988*; *Subbaiah et al., 2003*; *Ohvo et al., 1997*). These observations prompted an analysis of the relationship between fALOD4 binding and the lipid composition of the RBC membrane.

### fALOD4 binding is related to the phospholipid composition of the RBC membrane

To examine the lipid composition of the RBC membranes, we collected blood samples from a separate group of 41 African-Americans and 82 Hispanics who participated in the Dallas Biobank. The demographics and the distribution of fALOD4 binding in these individuals are shown in *Table 4*; the median fALOD4 binding was higher in Blacks than in Hispanics, which is similar to what we had seen in the other cohort (*Figure 5*). Lipids were extracted from RBC ghosts and then the major glycerophospholipids were quantitated by LC-MS/MS in the Kansas Lipidomics Research Center. The overall

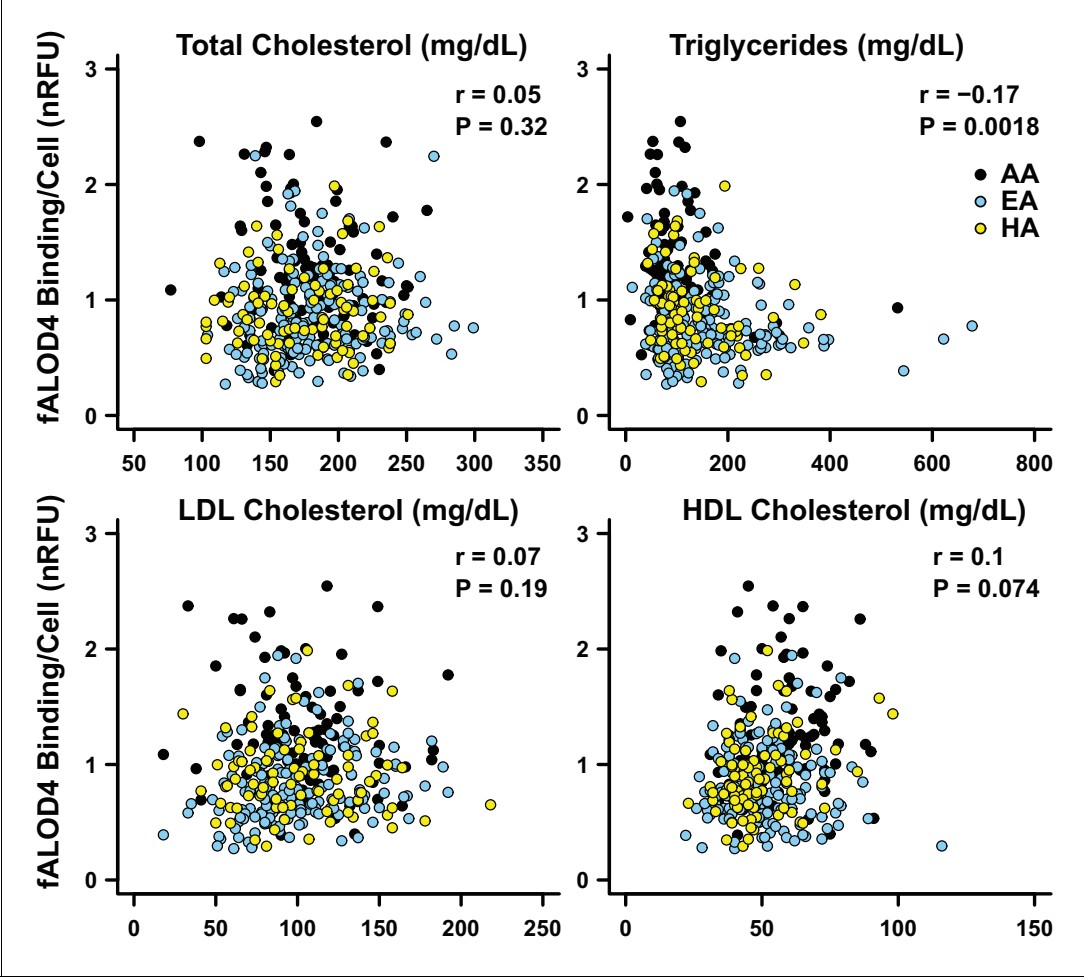

**Figure 7.** Correlations between fALOD4 binding to RBCs and serum lipids and lipoproteins. Total cholesterol, triglycerides, LDL-C and HDL-C were measured by Quest Diagnostics. fALOD4 binding to RBCs was measured in triplicate as described in the Materials and methods and normalized to the reference blood sample. Partial correlation coefficients (r) and p-values were calculated using linear regression adjusted for age, gender, and ethnicity, using statistical analysis software R.

lipid composition of the RBC membranes was similar to what has been reported previously (*Table 4*) (*Leidl et al., 2008*). Differences in levels of several of the RBC lipids were observed between Blacks and Hispanics, however in a multivariable models adjusted for age, gender, ethnicity, and plasma TG levels, only differences in lysophosphatidylethanolamine (LPE), SM, and phosphatidic acid levels were associated with differences in fALOD4 binding. Individually, these lipids accounted for 8.55% (LPE), 0.01% (SM), and 3.62% (phosphatidic acid) of variation in fALOD4 binding, respectively (*Table 5*). To determine the total contribution of the three lipid classes, a separate model was built that was adjusted for age, gender, ethnicity and plasma TG levels. In this model the RBC lipids accounted for 17.6% of the inter-individual variation in fALOD4 binding (the $R^2$ values are not additive because of the correlations among the RBC lipids).

We also examined the effect of differences in fatty acid composition of the lipids in the RBC membranes. None of the fatty acids had a major effect on RBC accessible cholesterol (*Table 5—source data 1*).

## Patients on hemodialysis have increased fALOD4 binding

Patients on hemodialysis have accelerated atherosclerosis that is not accounted for by known risk factors (*Levey et al., 1998*). To determine whether an RBC cholesterol activity is altered in these individuals, we measured fALOD4 binding to RBCs from 50 Black, White and Hispanic patients

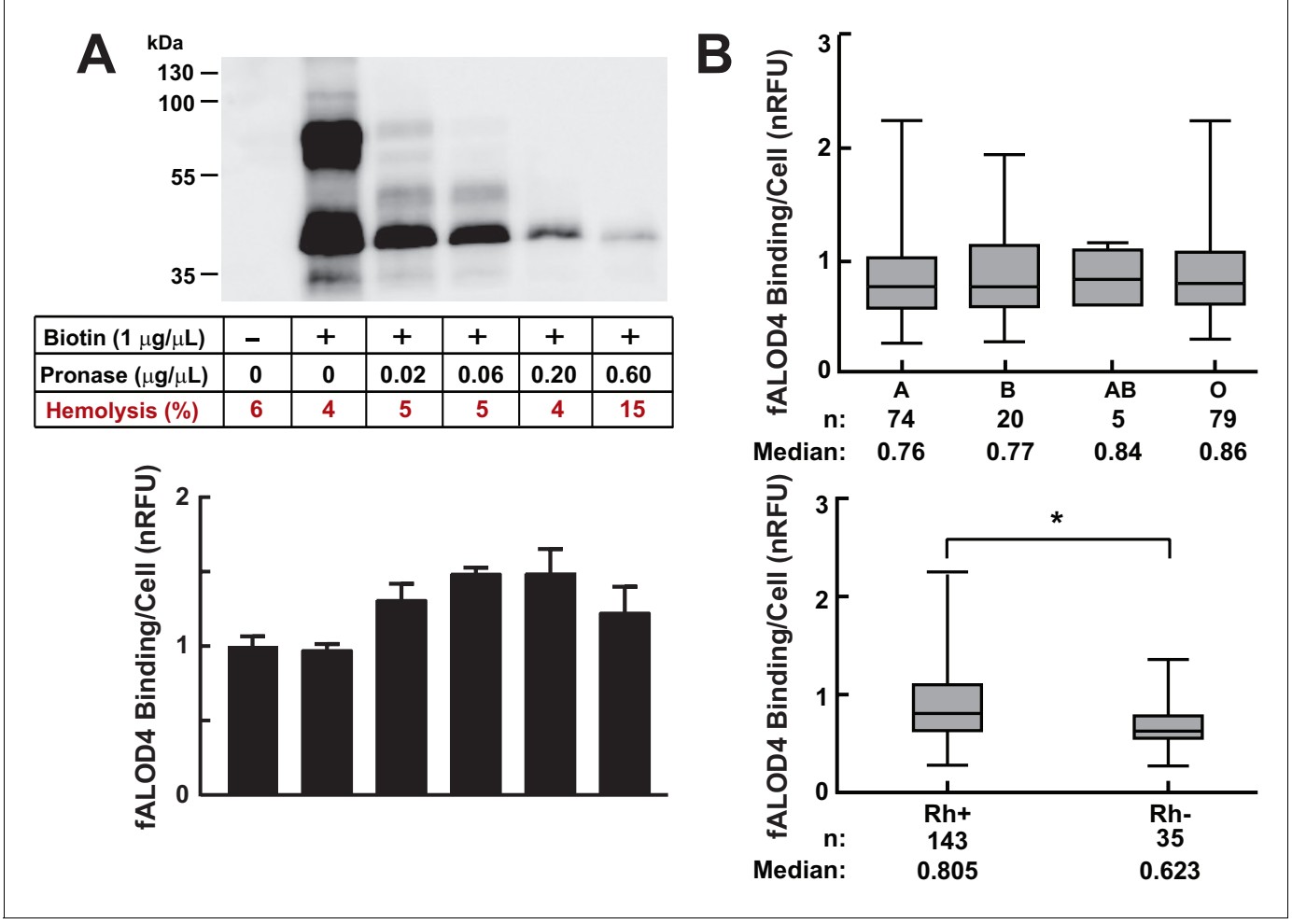

**Figure 8.** Effects of cell surface proteins on fALOD4 binding to RBCs. (**A**) Effect of proteolysis of RBC surface membrane proteins on fALOD4 binding to RBCs. RBCs isolated from a healthy individual were labeled with biotin and then treated with increasing amounts of pronase as indicated. An aliquot of the treated RBCs was used for fALOD4 binding assays and the remainder was subjected to 10% SDS/PAGE and probed with streptavidin-HRP (0.22 µg/mL) as described in the Materials and methods. fALOD4 binding values were normalized to the untreated sample. Data points represent the mean of three independent measurements. Error bars represent the SEM. The experiment was repeated three times and the results were similar. (**B**) Relationship between fALOD4 binding and ABO blood group antigens (upper panel) and Rhesus blood group (Rh antigen) (lower panel). RBC fALOD4 binding was determined in triplicate using RBCs from 178 White blood donors as described in the Materials and methods. Each fALOD4 binding measurement was performed in triplicate and values were normalized to the reference blood sample. Boxes represent the 25th and 75th percentiles and whiskers represent the minimum and maximum measurements. *p<0.005, two-tailed t-test. nRFU, normalized relative fluorescence units.

attending a hemodialysis clinic and in 44 ethnicity-matched controls measured on the same days (*Table 6*). fALOD4 binding was significantly higher in RBCs from dialysis patients that in ethnically-matched controls, irrespective of ethnicity (*Figure 10A*). In eight individuals, fALOD4 binding was measured in RBCs prepared from blood samples obtained before and after a single dialysis treatment. fALOD4 binding was similar in the paired samples (*Figure 10B*). Therefore, the increase in fALOD4 binding observed in dialysis patients was not an acute result of dialysis per se.

## Discussion

Here, we describe a new assay that measures the relative amount of accessible cholesterol in the membranes of RBCs. The assay is highly reproducible and captures a new quantitative trait in humans. The major finding of the study is that the amount of accessible cholesterol in RBC membranes is stable within an individual but varies >10 fold among individuals. The high degree of inter-

**Table 3.** Demographics and clinical characteristics of blood donors. Blood samples were obtained from healthy subjects after fasting for at least 8 hr. ABO blood group and presence of Rh antigen was determined by Carter BloodCare as described in Materials and methods. Ethnicity, age, and gender were self-reported. Plasma lipids and lipoproteins and RBC indices were measured by Quest Diagnostics as described in the Materials and methods. Numerical values are mean ± SD. Rh, rhesus antigen D; HDL-C, high density lipoprotein cholesterol; LDL-C, low density lipoprotein cholesterol; TG, triglyceride; Hb, hemoglobin; Hct, hematocrit; MCV, mean corpuscular volume.

| | |
|---|---|
| n | 178 |
| Age (mean) | 42 |
| Male (%) | 51 |
| Female (%) | 49 |
| Blood Groups | |
| A (%) | 42 |
| B (%) | 11 |
| AB (%) | 3 |
| O (%) | 445 |
| Rh positive (%) | 80 |
| Rh negative (%) | 20 |
| Total cholesterol (mg/dL) | 180 ± 3 |
| HDL-C (mg/dL) | 50 ± 1 |
| LDL-C (mg/dL) | 99 ± 3 |
| TG (mg/dL) | 155 ± 8 |
| Hb (g/dL) | 13.5 ± 0.2 |
| Hct (%) | 41.4 ± 0.5 |
| MCV (fL) | 88 ± 0.6 |
| RBC count ($10^6$/uL) | 4.7 ± 0.05 |

individual variation was unrelated to total cholesterol content of the RBC membrane and modestly affected by pronase-sensitive changes in protein content of the RBC membrane. Differences in RBC phospholipid composition of the RBC membrane accounted for 17.6% of the inter-individual variability (*Table 5*). Cholesterol accessibility in RBCs varied significantly between ethnic groups and correlated indirectly with plasma levels of TG (*Figure 7*), but not with the morphology or hemoglobin content of the RBCs (*Figure 6*). Thus, we have shown that the level of accessible cholesterol in RBCs in the population varies significantly among individuals in a stable fashion and we have identified several factors, both intrinsic and extrinsic to the RBC membrane, that contribute to the variation in RBC cholesterol accessibility. The significance of this variability on the trafficking of cholesterol among plasma components and tissues or if it relates to human disease awaits further study.

Among individuals, relative fALOD4 binding to RBCs did not correlate with the total cholesterol content of the RBC membranes (*Figure 4C*). Thus, the cholesterol in the RBC membrane that is accessible to fALOD4 represents a distinct pool (or state) of cholesterol, which is not reflected by direct assays of RBC cholesterol mass. The striking reproducibility of relative fALOD4 binding to RBCs within a given individual (*Figure 3*) indicates that the factors that govern RBC cholesterol accessibility must be relatively stable.

The lack of a relationship between cholesterol content and cholesterol accessibility in RBCs is similar to what has been observed among membranes from different organelles in mammalian cells, For example, plasma membranes (PMs) have a cholesterol content of ~40 mole%, whereas the cholesterol content of the membranes from the endoplasmic reticulum (ER) is only ~5 mole% (*Sokolov and Radhakrishnan, 2010*). Despite this wide difference in cholesterol content, the chemical activity of cholesterol in these two membranes may be similar (*Radhakrishnan and McConnell, 2000*). This

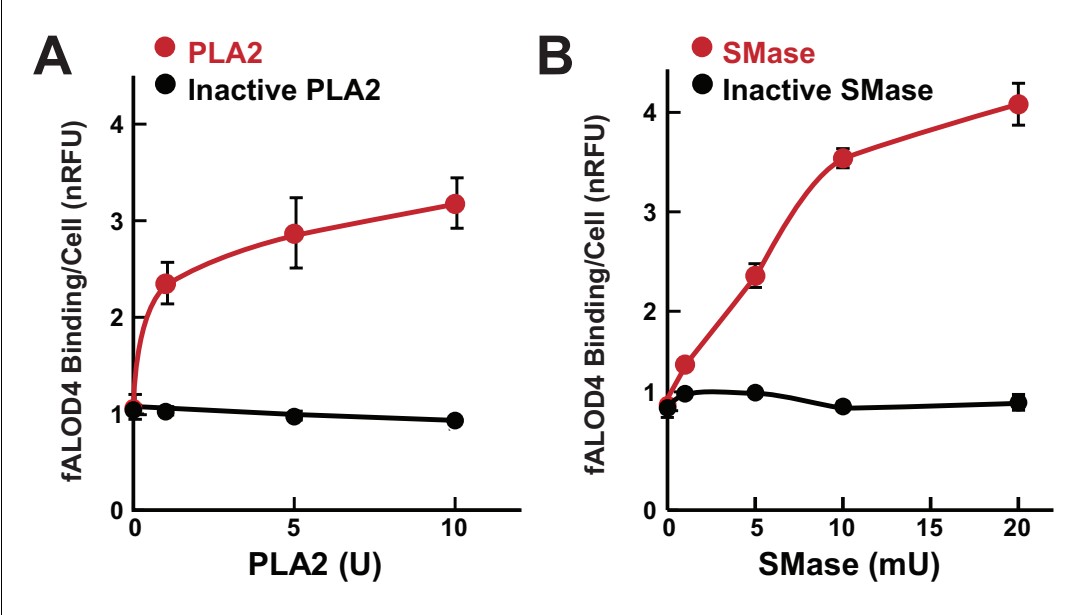

**Figure 9.** Effect of phospholipase and sphingomyelinase treatment of RBCs on fALOD4 binding. A total of $1.6 \times 10^8$ RBCs were incubated with indicated the amounts of active or inactivated phospholipase A2 (PLA2) (**A**) or active or inactive sphingomyelinase (SMase) (**B**) for 1 hr at room temperature or 37°C, respectively. Cells were washed three times in PBS and an aliquot containing $2 \times 10^5$ RBCs was used to measure fALOD4 binding. Data points represent the mean of three independent measurements (±SEM). The experiment was repeated three times and the results were similar.

equivalence could be achieved if PM phospholipids interact more strongly with cholesterol than ER phospholipids, a plausible possibility based on extensive fractionation analysis of PM and ER lipids (*Colbeau et al., 1971*; *Comte et al., 1976*; *Schroeder et al., 1976*) and biophysical studies of the varying affinity of cholesterol for different phospholipids (*McConnell and Radhakrishnan, 2003*; *Wattenberg and Silbert, 1983*; *Finean, 1953*). Fluctuations in membrane composition lead to changes in the chemical activity of cholesterol and could result in cholesterol transport between membranes, either internally or externally. In the case of the RBC, which has no internal organelle membranes, changes in cholesterol activity in the plasma membrane must equilibrate with cholesterol in lipoproteins, other plasma proteins, or with other cells, such as circulating leukocytes or endothelial cells (*Lange and Steck, 2016*).

We found that treatment of RBCs with either PLA2 or SMase increased cholesterol accessibility (*Figure 9*), a finding that is consistent with prior studies (*Subbaiah et al., 2003*; *Slotte et al., 1989*). Non-cholesterol lipids contributed significantly to inter-individual differences in cholesterol accessibility (*Table 4*), but identifying exactly which lipid species contributed to the observed variation in cholesterol accessibility was not straightforward. The complex population of phospholipids in the RBC membrane and the high degree of correlation between different phospholipid classes confounded the analysis. In the multivariate model adjusted for age, sex, lipid and lipoprotein levels, only LPE, SM and PA were associated with fALOD4 binding. Taken together, differences in these membrane lipids accounted for 17.6% of the variability in fALOD4 binding. It is also possible that differences in phospholipid composition alter the distribution of cholesterol between the two leaflets of the RBC membrane. Further studies in which the RBC membrane content of individual phospholipids can be selectively altered will be required to elucidate the effect of membrane phospholipids on cholesterol accessibility.

Why should RBC cholesterol accessibility be maintained at levels that closely bracket the threshold point of cholesterol accessibility? At cholesterol concentrations below the inflection point, the capacity of the membrane to incorporate cholesterol into complexes with phospholipids is high and the propensity for cholesterol to leave the membrane is low, therefore cholesterol will tend to accumulate in the RBC. At cholesterol concentrations above the inflection point, the capacity of the membrane to accommodate additional cholesterol in complexes with phospholipids is low, and

**Table 4.** Characteristics of 123 subjects in whom RBC membrane lipids were measured. Data are mean ± SD or median (25th – 75th percentile), unless otherwise indicated. All RBC lipids are expressed as mol% of total phospholipids. The lipid mole percentages were determined by mass tandem mass spectrometry. Continuous variables were compared between groups using ANOVA adjusted for age and gender where appropriate, categorical variables - using chi-square tests. Variables with non-normal distributions were inverse-normally transformed before analysis. HDL-C, high density lipoprotein cholesterol; LDL-C, low density lipoprotein cholesterol; TG, triglyceride; PC, phosphatidylcholine; SM, sphingomyelin; PE, phosphatidylethanolamine; ePC, ether-phosphatidylcholine; PS, phosphatidylserine; LPE, lysophosphatidylethanolamine; ePE, ether-phosphatidylethanolamine; LPC, lysophosphatidylcholine; PA, phosphatidic acid.

| Characteristic | Black (n = 41) | Hispanic (n = 82) | P-value |
|---|---|---|---|
| Age, years | 42.2 ± 11.5 | 34.8 ± 9.5 | 0.00087 |
| Female, n (%) | 34 (82.9) | 55 (67.1) | 0.10 |
| Male, n (%) | 7 (17.1) | 27 (32.9) | 0.10 |
| fALOD4 binding (nRFU) | 1.4 (1.1–1.7) | 1.7 (1.3–2.0) | 1.3 (1.0–1.5) |
| Total cholesterol (mg/dL) | 184.5 ± 38.5 | 185.2 ± 35.2 | 0.097 |
| HDL-C (mg/dL) | 59.7 ± 14 | 47.9 ± 11.8 | 0.00039 |
| LDL-C (mg/dL) | 107.5 ± 35.6 | 105.4 ± 29.6 | 0.35 |
| TG (mg/dL) | 71 (56–101) | 130 (89–190) | 1.2E-08 |
| *RBC membrane lipids* | | | |
| PC (mol %) | 40 ± 2.4 | 44 ± 3.6 | 2.7E-11 |
| SM (mol %) | 19 ± 4.1 | 19 ± 3.5 | 18 ± 4.4 |
| PE (mol %) | 15 ± 1.8 | 12 ± 1.8 | 6.6E-07 |
| ePC (mol %) | 5.7 ± 0.61 | 6.1 ± 0.92 | 0.0028 |
| PS (mol %) | 6.0 ± 0.94 | 6.2 ± 1.1 | 5.9 ± 0.81 |
| LPE (mol %) | 5.1 ± 1 | 4 ± 1.4 | 0.0027 |
| Ceramides (mol %) | 2.9 ± 0.45 | 2.8 ± 0.57 | 0.61 |
| PI (mol %) | 2.7 ± 0.63 | 2.7 ± 0.8 | 0.18 |
| ePE (mol %) | 1.7 ± 0.35 | 1.3 ± 0.26 | 8.3E-06 |
| LPC (mol %) | 1.4 ± 0.31 | 1.2 ± 0.29 | 2.3E-08 |
| PA | 0.44 ± 0.13 | 0.38 ± 0.099 | 0.0024 |

cholesterol is poised to leave the RBC membrane when suitable acceptors become available. Thus, the cholesterol content of the RBC membranes is maintained close to equilibrium with respect to cholesterol exchange.

It remains possible that differences in the complement of proteins in RBC membranes contribute to the observed inter-individual variability in fALOD4 binding. However, depletion of membrane proteins by pronase digestion had only a modest effect on fALOD4 binding (*Figure 8A*). We cannot exclude the possibility that in our experiments pronase treatment failed to digest a protein that contributes to the inter-individual differences in cholesterol accessibility. No association was found between ABO blood groups and fALOD4 binding (*Figure 8B*, upper panel), but fALOD4 binding was reproducibly lower in those who were negative for the Rh antigen, a transmembrane protein of unknown function (*Huang et al., 2000*) (*Figure 8B*, lower panel). The effect of the Rh antigen on fALOD4 binding may be direct or indirect. Rh antigen may compete with or directly inhibit fALOD4 binding. Alternatively, the antigen may alter the cholesterol accessibility in the RBC membrane. More extensive studies of the relationship between Rh antigenicity and cholesterol accessibility will be needed to elucidate the basis for this relationship.

**Table 5.** Factors associated with fALOD4 binding in subjects from **Table 4**. Beta coefficients are in SD units. For quantitative variables (age, lipids and CBCs), betas are given per 1 SD change in the predictor. Partial $R^2$ indicates the proportion of variance in fALOD4 binding explained by a given predictor after adjusting for other factors. LPE, lysophosphatidylethanolamine; SM, sphingomyelin; PA, phosphatidic acid.

| Factor | Beta | SE | P-value | Partial $R^2$ (%) |
|---|---|---|---|---|
| Age | −0.044 | 0.034 | 0.20 | 1.40 |
| Male Gender | −0.008 | 0.068 | 0.91 | 0.01 |
| Ethnicity (AA vs. HA) | −0.442 | 0.077 | 6.9E-08 | 22.44 |
| Triglycerides | −0.192 | 0.033 | 4.6E-08 | 22.96 |
| LPE | −0.131 | 0.040 | 0.0014 | 8.55 |
| SM | −0.005 | 0.045 | 0.91 | 0.01 |
| PA | 0.076 | 0.036 | 0.040 | 3.62 |
| Total | | | | 56.25 |

**Source data 1.** Correlation between fatty acids (FA) and fALOD4 binding to RBCs.

Factors that are extrinsic to the RBC were also found to be associated with differences in RBC cholesterol accessibility. Lipoprotein lipids accounted for 5.4% of the variation in fALOD4 binding in a multivariate model that included gender and race (**Table 2**). Plasma levels of TG and RBC cholesterol accessibility were inversely correlated. This correlation may be a direct result of movement of accessible cholesterol from RBCs to very low density lipoproteins (VLDL) (**Chung et al., 1986**). It is also possible that circulating lipoproteins influence RBC cholesterol accessibility by transferring phospholipids to (or from) the RBC membrane (**Reed, 1968**).

**Table 6.** Demographics and lipid levels of 50 subjects with chronic renal insufficiency on hemodialysis and 44 ethnically-match controls. Data are mean ± SD, unless otherwise indicated. Blood samples were obtained either from hemodialysis patients immediately prior to dialysis or from healthy subjects after a minimum of 8 hr of fasting. Ethnicity, age, and gender were self-reported. Plasma lipids and lipoproteins and RBC indices were measured by Quest Diagnostics as described in the Materials and methods. HDL-C, high density lipoprotein cholesterol; LDL-C, low density lipoprotein cholesterol; TG, triglyceride; Hb, hemoglobin; Hct, hematocrit; MCV, mean corpuscular volume.

| | All | | White | | Black | | Hispanic | |
|---|---|---|---|---|---|---|---|---|
| | Control | Dialysis | Control | Dialysis | Control | Dialysis | Control | Dialysis |
| N | 44 | 50 | 26 | 8 | 10 | 32 | 8 | 10 |
| Age | 44 ± 2 | 54 ± 2 | 48 ± 2 | 52 ± 6 | 51 ± 3 | 57 ± 2 | 27 ± 4 | 48 ± 4 |
| Male (%) | 46 | 58 | 54 | 71 | 30 | 50 | 56 | 80 |
| Total cholesterol (mg/dL) | 182 ± 6 | 142 ± 5 | 191 ± 8 | 180 ± 3 | 153 ± 12 | 156 ± 6 | 184 ± 13 | 116 ± 5\5 |
| HDL-C (mg/dL) | 48 ± 2 | 45 ± 2 | 45 ± 3 | 40 ± 4 | 54 ± 4 | 49 ± 2 | 48 ± 3 | 38 ± 4 |
| LDL-C (mg/dL) | 103 ± 6 | 72 ± 4 | 111 ± 7 | 84 ± 15 | 83 ± 10 | 75 ± 5 | 100 ± 7 | 54 ± 4 |
| TG (mg/dL) | 153 ± 15 | 122 ± 11 | 172 ± 22 | 144 ± 20 | 78 ± 9 | 109 ± 15 | 180 ± 30 | 147 ± 22 |
| Hb (g/dL) | 13.8 ± 0.2 | 11.8 ± 0.8 | 14 ± 0.3 | 16 ± 3 | 13 ± 0.5 | 10.4 ± 0.4 | 13.5 ± 0.4 | 12.6 ± 2 |
| Hct (%) | 42 ± 1 | 32 ± 1 | 43 ± 1 | 31 ± 1 | 41 ± 2 | 32 ± 1 | 41 ± 1.2 | 31 ± 1 |
| MCV (fL) | 87 ± 1 | 90 ± 1 | 86 ± 2 | 87 ± 6 | 90 ± 2 | 92 ± 1 | 86 ± 2 | 88 ± 4 |
| RBC count ($10^6$/µL) | 4.8 ± 0.1 | 3.7 ± 0.2 | 4.8 ± 0.2 | 3.4 ± 0.1 | 4.7 ± 0.3 | 3.8 ± 0.3 | 4.8 ± 0.2 | 3.5 ± 0.1 |

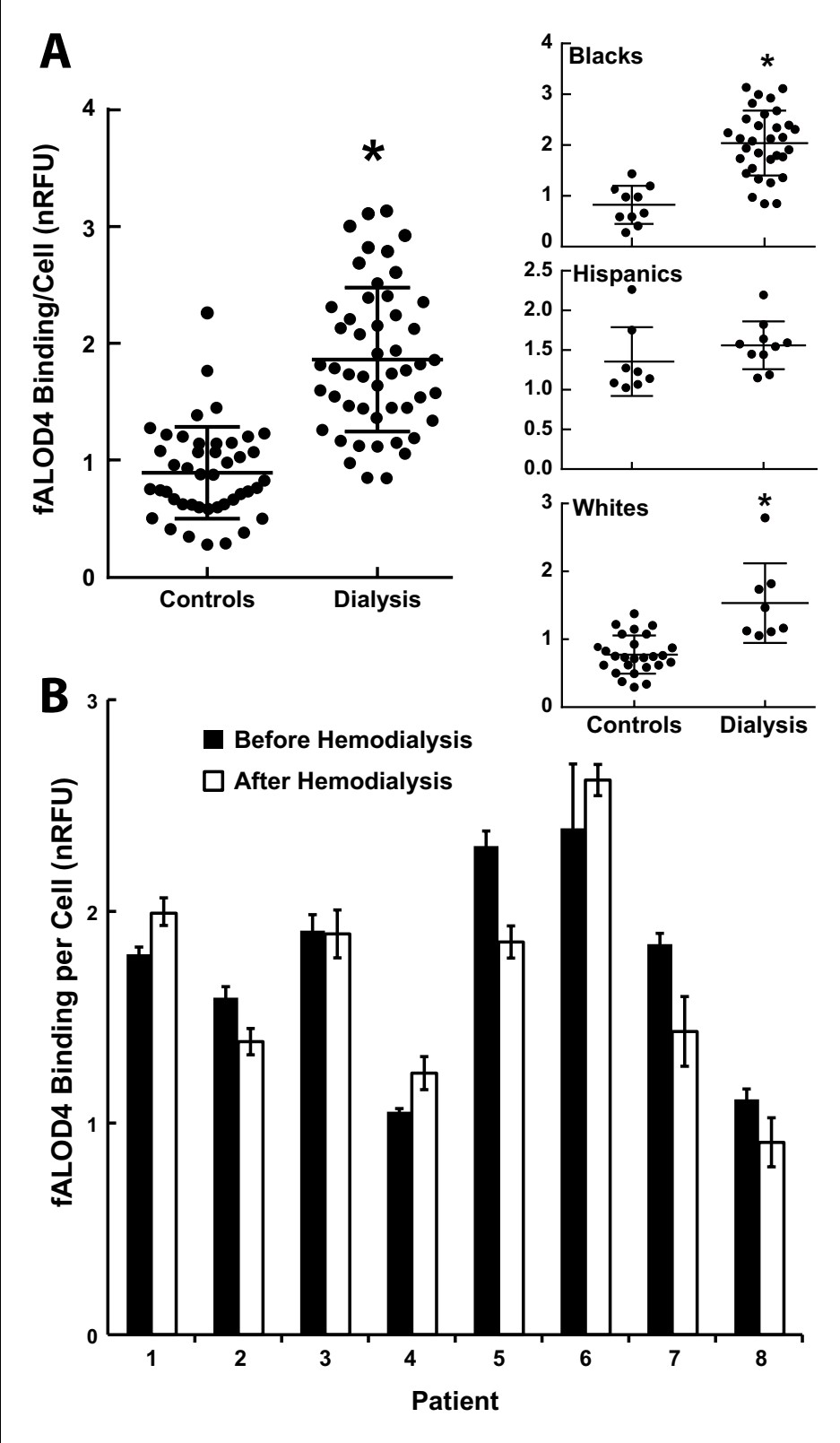

**Figure 10.** fALOD4 binding to RBCs of patients on hemodialysis. (**A**) Blood samples were collected from 50 hemodialysis patients immediately before dialysis was initiated and from 44 ethnicity-matched individuals who did not have chronic renal failure and were measured on the same day (*Table 6*). RBCs were isolated and fALOD4 binding assays were performed as described in the Materials and methods. Each measurement was performed in

*Figure 10 continued*

triplicate. fALOD4 binding values were normalized to the binding values obtained for the reference blood sample that was collected from the same healthy subject for each experiment. Data is plotted as the mean of each nomalized value. *p<0.0001, two-tailed t-test. (B) fALOD4 binding measurements were made exactly as described above using RBCs isolated from blood samples collected before and after dialysis in eight individuals undergoing hemodialysis. Blood samples were collected once from each patient before and after hemodialysis in eight patients. Data is plotted as the mean ± SEM.

It is possible that ABCA1 transfers cholesterol directly from cells to RBCs, and thus contributes to the flux of cholesterol from extrahepatic tissues to the liver (*Figure 1*). RBCs do not have access to peripheral tissues unless there is loss of vascular integrity, but they can interact indirectly with peripheral tissues through cells, such as macrophages, and lipoproteins that enter the tissues and return to the circulation. RBCs have been shown to be excellent acceptors of free cholesterol from macrophages (*Ho et al., 1980*). Brown and Goldstein found that RBCs were 6-fold more potent than whole serum as cholesterol acceptors from macrophages (*Ho et al., 1980*). Lipoprotein-depleted plasma is as efficient an acceptor as whole serum or plasma from cholesterol-loaded macrophages (*Ho et al., 1980*). Although here we focused on the relationship between RBC accessible cholesterol and lipoprotein levels, it is also possible that blood components other than lipoproteins contribute to differences in cholesterol accessibility of the RBC membrane (*Ho et al., 1980*). RBCs may also exchange cholesterol with endothelial cells, thus indirectly transporting cholesterol from the tissues. It is unlikely that RBCs deliver cholesterol directly to hepatocytes since they are too large to enter the space of Disse.

Significant and systematic differences in RBC cholesterol activity were found among individuals of European, African, and Hispanic descent. Median fALOD4 binding was highest in African-Americans (1.18 nRFU), intermediate in Hispanics (0.86 nRFU) and lowest in European-Americans (0.77 nRFU) (*Figure 5*). These differences were seen in two independent samples collected at different times (*Figure 5* and *Table 4*). The increased RBC accessibility in Blacks was not fully explained by the measured differences in RBC lipid composition or by the plasma lipoprotein levels. In linear regression analysis, adjusted for these factors, the proportion of variability in fALOD4 binding explained by ethnicity was reduced from 32% to 22.4%, but remained highly statistically significant (*Table 5*). Dietary differences between the ethnic groups may also contribute to differences in RBC cholesterol accessibility. It would be informative to measure fALOD4 binding to RBCs in a larger multiethnic sample and use genetic association to identify potential factors accounting for the observed differences between both individuals and ethnic groups.

Interethnic differences in cholesterol accessibility may be the consequence(s) of historic differences in selective pressure among the three ethnic groups. RBCs with higher fALOD4 binding would be anticipated to be more susceptible to microorganisms that produce cholesterol-binding toxins. ALO, produced by *Bacillus Anthracis*, is a member of the cholesterol-dependent cytolysin family, which is a family of pore-forming toxins produced by a myriad of different gram positive bacteria, including species of *Streptococcus*, *Listeria* and *Clostridium* (*Palmer, 2001*). Perhaps individuals with lower fALOD4 binding are less susceptible to the hemolytic effects of these toxins, thus providing them with a survival advantage.

Genetic and pharmacological studies have provided compelling evidence that increased circulating levels of HDL-cholesterol, widely viewed as the vehicle for reverse cholesterol transport, do not confer protection against coronary heart disease (*Siddiqi et al., 2015*). RBCs may be a conduit for transporting cholesterol from peripheral tissues to the liver. Mice made anemic had reduced fecal excretion of cholesterol from macrophages (*Hung et al., 2012*), and anemia has been shown to be associated with increased coronary artery disease (Hazard Ratio of 1.41 over 6 years) (*Sarnak et al., 2002*). To determine if RBC cholesterol accessibility provides a clinically relevant biomarker, we measured cholesterol accessibility in patients with chronic renal failure on dialysis, who have a dramatically increased risk of atherosclerosis that is not explained by known risk factors (*Levey et al., 1998*). We found a systematic increase in cholesterol accessibility in RBCs in patients on dialysis when compared to healthy age- and race-matched controls. The cause of these differences and whether they

are related to the increase in atherosclerosis seen in this population is not known at this time and will require further study.

## Materials and methods

### Materials

Methyl-$\beta$-cyclodextrin (MCD) and hydroxypropyl-$\beta$-cyclodextrin (HPCD) were obtained from the Cyclodextrin Technologies Development, Inc. (Alachua, FL). Dithiothreitol (DTT) and pronase (Cat. No. 10165921001) were purchased from Roche (Indianapolis, IN). Phospholipase A2 (Cat. No. P9279), sphingomyelinase (SMase) (Cat. No. S8633), and Tris (2-carboxyethyl) phosphine (TCEP) were purchased from Sigma-Aldrich (St. Louis, MO). Alexa Fluor 488 $C_5$-maleimide, EZ-Link Sulfo-NHS-LC-Biotin, Pierce High Sensitivity Streptavidin-HRP and SuperSignal West Pico Chemilumines-cent substrate were purchased from Thermo Fisher Scientific (Waltham, MA). Newborn calf lipopro-tein-deficient serum (NCLPPS) was prepared by ultracentrifugation as previously described (*Goldstein et al., 1983*). Stock solutions of cholesterol/MCD complexes were prepared at a final cholesterol concentration of 2.5 mM and a cholesterol: MCD molar ratio of 1:10 as described (*Brown et al., 2002*).

Buffer A contained 50 mM Tris-HCl (pH 7.5) and 1 mM TCEP; buffer B contained 50 mM Tris-HCl (pH 7.5), 150 mM NaCl, and 1 mM DTT; buffer C contained 50 mM Tris-HCl (pH 7.5), 200 mM NaCl, and 12% (v/v) glycerol; buffer D contained phosphate buffered saline (PBS) supplemented with 1 mM EDTA and 2% (v/v) NCLPPS; buffer E contained 10 mM Tris-HCl (pH 7.5); Buffer F contains 50 mM Tris-HCl (pH 8.0), 100 mM NaCl, 0.5% (w/v) SDS, and 1 mM DTT.

### Protein purification and labeling with fluorescent dye

A pRSETB expression vector (Life Technologies, Carlsbad, CA) that encodes domain 4 (amino acids 404–512) of anthrolysin O (ALO) (ALOD4) from *Bacillus Anthracis* (*Bourdeau et al., 2009*) was used for these studies. The construct contains two point mutations (S404C and C472A) and has a His$_6$ tag at the NH$_2$-terminus (*Gay et al., 2015*). Recombinant ALOD4 was purified from *E. coli* expressing ALOD4 using nickel chromatography as described (*Gay et al., 2015*). Typical yields from a 6L bacte-rial culture were ~10–15 mg. Purified ALOD4 was concentrated to ~0.8 mg/mL (final volume 15 mL) using a 10,000 MWCO Amicon Ultra Centrifugal Filter (EMD Millipore. Billerica, MA) and mixed with 135 mL of buffer A to lower the salt concentration to a final value of 10 mM NaCl. The mixture was applied to a 1 mL HiTrap Q Sepharose High Performance column (GE Healthcare, Little Chalfont, UK) that had been pre-equilibrated in buffer B. After the column was washed with 100 column vol-umes of buffer A and 10 column volumes of buffer A plus 50 mM NaCl, ALOD4 was eluted with NaCl into a single 2 mL fraction using buffer A containing 500 mM NaCl. The final concentration of ALOD4 (MW, 16.2 kDa) was between 6 and 10 mg/mL.

Each labeling reaction contained 200 nmoles ALOD4 and 1400 nmoles Alexa Fluor 488 $C_5$-malei-mide dye in buffer B (1 ml) with 250 mM NaCl. After incubation for 16 hr at 4°C, unincorporated dye was removed by passing the sample over Ni-NTA agarose beads (1 mL) pre-equilibrated with buffer B. The column was washed with 10 column volumes of buffer B containing 50 mM imidazole. Bound proteins were eluted with buffer B containing 300 mM imidazole. Fractions were subjected to SDS-PAGE followed by Coomassie Brilliant Blue R-250 staining and those containing the recombinant protein were pooled and concentrated by anion exchange chromatography, as described above. Fluorescently labeled protein, referred to as fALOD4, was diluted to a final concentration of 30 μM in buffer C, aliquoted, flash frozen, and stored at −80°C. fALOD4 concentrations were measured using the Pierce 660 nm Protein Assay Reagent (Thermo Fisher Scientific). Alexa Fluor 488 dye con-centrations were measured with a NanoDrop instrument (Thermos Scientific, Wilmington, DE) using the extinction coefficient of Alexa Fluor 488 ($\varepsilon_{495}$ = 76,000 M$^{-1}$ cm$^{-1}$). The degree of labeling of fALOD4 by Alexa Fluor 488 ranged from 0.64–0.95.

### Human subjects and red blood cell isolation

The study was performed following the guidelines of the local medical ethical committee and in accordance with the declaration of Helsinki. The study protocol was approved by the Institutional Review Board of UT Southwestern Medical Center. Written informed consent was obtained from all

blood donors prior to participation in this study. Each participant completed a detailed staff-administered survey, including questions about socioeconomic status, medical history and medication use. Ancestry was self-reported. Venous blood samples (7 ml) were obtained from individuals who had fasted at least 8 hr. Blood samples from hemodialysis patients were obtained immediately prior to dialysis. Blood was collected in citrate-EDTA containing tubes (BD Medical Supplies, Franklin Lakes, NJ). Each study participant completed a detailed survey to determine weight, gender, BMI, socio-economic status, medical history, and current and past medication use. Race and ethnicity were self-reported. A complete blood count [red blood cell (RBC) count, hemoglobin (Hb), hematocrit (Hct), mean corpuscular volume (MCV), mean corpuscular hemoglobin concentration (MCHC), white blood cell (WBC) count, platelets] and measurement of plasma levels of lipids (cholesterol and triglyceride) and lipoproteins were performed by Quest Diagnostics. Blood typing for ABO and Rhesus antigen (D antigen) was performed at Carter Bloodcare in Dallas, TX for the blood samples that were obtained from that organization.

For isolation of the RBCs, the blood samples were maintained at 4°C after collection. Within 3 hr of collection, the samples were subjected to centrifugation at 1500 x $g$ for 10 min at 4°C. Plasma was stored at −80°C and the RBCs were resuspended in 4 mL ice-cold PBS. The RBCs in PBS were stored at 4°C until used for experiments, which were performed within 5 days.

## Assay for fALOD4 binding to RBCs

To determine optimal conditions, 5 μL of RBCs (~2 × $10^7$ cells) was diluted in 495 μL ice-cold PBS with gentle mixing and then subjected to centrifugation at 2500 x $g$ for 2.5 min at 4°C. The pellet was resuspended in 500 μL ice-cold PBS, and the above procedure was repeated twice. The resulting pellet was resuspended in 500 μL ice-cold PBS. A total of 5 μL (~2.5×$10^5$ cells) were added to 490 μL of buffer D and 5 μL of buffer C containing the indicated amounts of fALOD4. After incubation for 3 hr at 4°C on a rotator, samples were subjected to centrifugation at 2500 x $g$ for 2.5 min at 4°C. The resulting supernatant was saved for analysis of released hemoglobin, and the pellet containing RBCs and bound fALOD4 was resuspended in 200 μL PBS and transferred to 12 × 75 mm round bottom polystyrene tubes (Falcon, Corning, NY) for flow cytometry analysis. In a second set of samples, fALOD4 binding assays were performed by diluting 10 μL of RBCs in 1 mL ice-cold PBS with gentle mixing and then centrifuged at 2500 x $g$ for 2.5 min at 4°C. The pellet was resuspended in 1 mL ice-cold PBS and the above washing procedure was repeated twice. A total of 10 μL of this washed dilution (~4 × $10^5$ RBCs) was combined with 485 μL ice-cold buffer D and 5 μL buffer C containing 30 μM fALOD4. The remainder of the protocol was carried out as described above.

## Flow cytometry analysis of fALOD4 binding to RBCs

All data were acquired using a FACSCalibur flow cytometer (BD Biosciences, San Jose, CA), equipped with an argon ion laser (488 nm) to detect Alexa Fluor 488 dye (excitation maximum: 493 nm; emission maximum: 516 nm) on RBC-bound fALOD4. CellQuest software (BD Biosciences) was used to set the detector for forward-scatter (FSC) in linear mode, and detectors for side-scatter (SSC) and FL1 (to monitor Alexa Fluor 488) to logarithmic mode. A total of 10,000 RBCs were analyzed for each experimental condition unless otherwise indicated. A control sample containing untreated RBCs was used to calibrate the detectors and to correct for auto-fluorescence of samples. The FSC/SSC profile of the control sample was used to gate around RBCs with normal morphology and to separate fluorescently labeled RBCS from unlabeled cells. Flow cytometry data analysis was conducted using FlowJo software version 9 (Ashland, OR). Median fluorescent intensity (MFI) of each population of fluorescently labeled RBCs was calculated and used to quantify fALOD4 binding per RBC.

The median Alexa Fluor 488 signal per RBC is hereafter referred to as fALOD4 binding per cell. fALOD4 binding values for each sample were normalized to the binding value obtained from a reference blood sample. The reference blood sample was collected from the same healthy subject and processed with the samples in each experiment.

## Modulation of RBC cholesterol

RBCs were isolated from whole blood as described above. In a typical experiment, one ml of RBCs was diluted in 9 mL of ice cold PBS and gently mixed and then subjected to centrifugation at 2500 x

*g* for 5 min at 4°C. The pellet was resuspended in 9 mL ice-cold PBS, and the above procedure was repeated twice. The resulting pellet was resuspended in 9 mL ice-cold PBS. A total of 1 mL RBCs (~4 × $10^8$ cells) were added to either 0–2% (w/v) HPCD (to deplete RBC cholesterol) or 0–75 μM cholesterol/MCD complexes (to increase RBC cholesterol) in a final volume of 2 mL ice-cold PBS. After incubation for 1 hr at 4°C on a rotator, samples were subjected to centrifugation at 2500 x *g* for 2.5 min at 4°C. The resulting supernatant was removed and saved for analysis of released hemoglobin. The RBC-containing pellets were washed three times in 1 mL of ice-cold PBS as described above. After the final wash, the pellet containing the RBCs was resuspended in 1 mL ice-cold PBS. An aliquot of the RBCs (10 μL of total, ~4 × $10^6$ RBCs) was diluted in 100 μL PBS and used for fALOD4 binding assays and flow cytometry analysis as described above. The remainder of RBCs was used for cholesterol quantification (see below).

## Preparation of RBC ghost membranes

RBC ghost membranes were prepared by a modified Dodge method (*Dodge, 1963*). The RBC suspension was centrifuged at 2500 x *g* for 2.5 min at 4°C. The supernatant was discarded and the pellet was hemolysed by resuspending the RBCs in 1 mL of hypotonic buffer E and subjecting the mixture to vigorous vortexing. The lysed RBCs were subjected to centrifugation at 13,100 x *g* for 15 min at room temperature and the supernatant was discarded. The pellet, which contained membranes from lysed RBCs and intact RBCs, was subjected to additional rounds of lysis and centrifugation until the 13,100 x *g* supernatant was clear. The pellet from the final centrifugation step, designated ghost membranes, was reconstituted in 500 μL PBS and stored at −20°C if not used immediately.

## Quantification of released hemoglobin (Hb)

Aliquots of 2500 x *g* supernatants (100 μL) generated during fALOD4 binding assays, RBC cholesterol modulation assays, and proteolytic cleavage assays, were transferred to 96 well clear bottom plates (Evergreen Scientific, Los Angeles, CA). Amounts of hemoglobin released from the RBCs during the assay were quantified using a microplate reader (BioTek, Winooski, VT) by measuring absorbance at 540 nm. As controls for hemolysis, an equal number of RBCs were mixed with 1% Triton-X detergent (0.5 mL final volume) or with PBS buffer alone. The samples were subjected to centrifugation (2500 *g* for 2.5 min) and 100 μl aliquots of the supernatant were transferred to 96-well plates. Percent of hemolysis was calculated as [Hb-absorbance of sample / Hb-absorbance of positive control]*100.

## Proteolytic cleavage of human RBC surface proteins

Cell surface proteins on RBC membranes were modified with biotin as described previously (*Ferru et al., 2012*). Each reaction contained EZ-Link Sulfo-NHS-LC-Biotin (0.5 mg) and 500 μl of washed RBCs (2 × $10^7$ RBCs) In a final volume of 1 mL ice-cold PBS. After a 30 min incubation at 4°C on a rotator, the RBCs were washed three times with PBS supplemented with 100 mM glycine to remove excess biotin. The biotin-treated RBCs were resuspended in 0.5 mL ice-cold PBS.

RBC surface proteins were digested by adding pronase (0–300 μg) to 130 μL of biotin-modified RBCs (~5 × $10^6$) in a final volume of 500 μL. After a 1 hr incubation at 40°C on a rotator, the samples were subjected to centrifugation at 2500 x *g* for 2.5 min at room temperature and the released hemoglobin was measured in the supernatants. RBC pellets were washed three times in PBS at room temperature and then resuspended in 130 μL PBS. An aliquot (15 μL) was used for fALOD4 binding assay and the remaining RBCs (115 μL) were subjected to centrifugation at 2500 x *g* for 2.5 min at room temperature. The pellet was resuspended in 40 μL of SDS-containing Buffer F and vortexed before being subjected to SDS-PAGE. The proteins were transferred to nitrocellulose membranes and then incubated in 1.25% milk in the absence and then in the presence of Pierce high affinity streptavidin-HRP (0.22 μg/mL) (Thermo Fisher Scientific) to detect the biotinylated proteins. Bound streptavidin-HRP was visualized after treatment of the membranes with SuperSignal West Pico Chemiluminescent substrate (Thermo Fischer Scientific) using an Odyssey FC Imaging machine (LI-COR Biosciences, Lincoln, NE).

## Analysis of RBC lipids

To measure cholesterol, ghost membranes were generated from ~$5 \times 10^8$ RBCs as described above. Lipids were extracted using the Folch extraction method (*Folch et al., 1957*). An aliquot of the extracted lipids (10% of total) was used to measure cholesterol by Infinity cholesterol reagent (Thermo Fisher Scientific) or by gas chromatography (*Wilund et al., 2004*). Another aliquot (10% of total) was used to measure choline-containing phospholipids with the Wako Phospholipids C enzymatic kit (Wako Diagnostics, Mountain View, CA). This value was divided by a calibration factor of 0.55 to estimate the total amount of RBC phospholipids. The calibration factor was derived from previous quantitative studies where it was determined that choline-containing phospholipids make up 55% of RBC phospholipids (*Dodge and Phillips, 1967*). Mole percentages of cholesterol were calculated by dividing moles of cholesterol by moles of total lipids (cholesterol plus phospholipids) and multiplying the result by 100. We used 389 Da for the molecular mass of cholesterol and 800 Da as an estimate for the average molecular mass of a phospholipid.

To measure RBC phospholipids, the ghost membranes were prepared as described above and lipids were extracted using a modified Bligh-Dyer extraction (1:1:1 mixture of dichloromethane: methanol: PBS) (*Folch et al., 1957*). In short, membranes were suspended in a volume of 3 mL PBS and added to an 8 mL, $13 \times 100$ mm glass culture tube (Fisherbrand, Pittsburgh, PA) containing 3 mL methanol that was immersed in a water bath sonicator (Branson model # B2510MT, Danbury, CT) set at maximum speed. After removal of the tube from the sonicator, a total of 3 mL of dichloromethane (DCM) was added to the tube. The samples were capped (Fisherbrand black phenolic Teflon coated screw cap), vortexed, and centrifuged at 2500 rpm for 5 min at room temperature. The bottom phase, which contained the lipids, was transferred to a fresh glass culture tube. An additional 4 mL DCM was added to the original tube, which was again capped, vortexed and centrifuged at 2500 rpm for 2.5 min at room temperature. The bottom phase was combined with the material from the initial phase separation (total volume of approximately 5 mL). The samples were placed at 42°C under a gentle stream of nitrogen gas until the volume was approximately 0.5 mL. The samples were transferred to 1 mL glass culture tubes, which were again placed at 42°C under a gentle stream of nitrogen gas until all solvent had evaporated. The glass tubes containing the dried lipids were immediately capped, flash frozen with liquid nitrogen, placed on dry ice, and stored at −80°C until they were shipped on dry ice to the Kansas Lipid Core for detailed analysis of polar lipids.

## Phospholipase A2 (PLA2) and sphingomyelinase (SMase) treatment of RBCs

Washed RBCs ($1.6 \times 10^8$ RBCs) in 300 µL were incubated with PLA2 (40 U/mL) (Sigma Aldrich, St. Louis, MO) in Ringers buffer (active PLA2) or divalent ion free PBS (inactive PLA2) in a final volume of 1 mL and incubated for 1 hr at room temperature. Cells were washed three times with 1 mL ice-cold PBS buffer and a fALOD4 binding assay was performed (~$2 \times 10^5$ RBCs).

A similar experiment was performed using neutral sphingomyelinase *from Staphylococcus aureus* (SMase) (0.195 mL in 0.25 M PBS containing 50% glycerol, pH 7.5) (Sigma Aldrich, St. Louis, MO). SMase was diluted in Ringers buffer (active enzyme) or in PBS buffer (inactive enzyme). Washed RBCs ($1.6 \times 10^8$ RBCs) were brought up to a final volume of 1 mL and incubated with SMase (0–20 mU/mL) for 1 hr at 37°C on a rotator. RBCs were washed three times and then fALOD4 binding assays were performed on 3 µL aliquots (~$2 \times 10^5$ RBCs) of SMase-treated cells. A similar experiment was performed using SMase (0.195 mL in 0.25 M PBS containing 50% glycerol, pH 7.5), which was diluted 100-fold in Ringers lactate (300 mL total volume). Reaction mixtures containing 300 µL washed RBCs ($1.6 \times 10^8$ RBCs) and 0–20 milliunits/mL of SMase in a final volume of 1 mL with Ringers buffer were incubated for 1 hr at 37°C on a rotator. RBCs were then washed three times with 1 mL ice-cold PBS buffer and a 3 µL aliquot ($1.6 \times 10^5$ RBCs) was used to measure fALOD4 binding as described above. The remainder of the SMAse treated RBCs were pooled and RBC ghost membranes were prepared. Protein and lipid analysis of ghost membranes was performed as described above.

## Statistical analysis

Data are presented as mean ± SEM, mean ± SD, or median (25th – 75th percentile), as indicated. Continuous characteristics were compared using t-tests or ANOVA adjusted for age and gender

where appropriate; categorical variables were compared using chi-square tests. Variables with non-normal distributions were logarithmically or inverse-normally transformed before analysis. The relationships between fALOD4 binding and lipid levels and RBC indices were assessed using linear regression adjusted for age, gender, and ethnicity. Multivariable-adjusted regression models were used to determine which factors jointly contributed to variability in fALOD4 binding. We used a forward selection procedure to build the fully adjusted model. Briefly, our baseline model included age, gender and ethnicity as covariates, to account for differences in age and gender distribution between the ethnic groups. Next, we added plasma lipid and lipoprotein levels, RBC indices, and RBC lipids to the baseline model, one at a time. The factor with the most significant contribution to the model (lowest p-value) was added to the model, and the procedure was repeated, until no more variables were significantly associated with fALOD4 binding. Partial $R^2$ values were calculated based on the final model, to assess the proportion of variance in fALOD4 binding explained by each factor. All statistical analyses were performed using the software package Prism 6 (GraphPad, La Jolla, CA) or R statistical analysis software version 3.2.3 (www.R-project.org/).

## Acknowledgements

We wish to thank Stephanie Spaeth for excellent technical assistance. We also thank Teresa Eversole and Barbara Gilbert for collection of the blood samples from study participants. We thank the Wadley Blood Bank at Carter BloodCare for contributing blood samples to this study. We also thank Drs. Miguel Vazquez and Tamim Hamdi for assisting in our collection of blood samples from patients on dialysis. The lipid profile data were acquired at the Kansas Lipidomics Research Center (KLRC). Instrument acquisition and method development at KLRC were supported by NSF grants MCB 0455318, MCB 0920663, DBI 0521587, DBI 1228622, Kansas INBRE (NIH Grant P20 RR16475 from the INBRE program of the National Center for Research Resources), NSF EPSCoR grant EPS-0236913, Kansas Technology Enterprise Corporation, and Kansas State University. Both Sally A. Ingham and Rima Shah Chakrabarti were supported by the Howard Hughes Medical Fellows Research Program. This work was supported by a grant from the National Institute of Medicine (PO1 HL20948, UL1TR001105).

## Additional information

### Competing interests

HHH: Reviewing editor, *eLife*. The other authors declare that no competing interests exist.

### Funding

| Funder | Grant reference number | Author |
| --- | --- | --- |
| Howard Hughes Medical Institute | | Rima S Chakrabarti<br>Sally A Ingham<br>Helen H Hobbs |
| National Institutes of Health | PO1 HL20948 | Austin Gay<br>Jonathan C Cohen<br>Arun Radhakrishnan<br>Helen H Hobbs |
| Welch Foundation | I-1793 | Arun Radhakrishnan |
| American Heart Association | 12SDG12040267 | Arun Radhakrishnan |
| National Institutes of Health | 5T32-GM008203 | Austin Gay |
| National Institutes of Health | UL1TR001105 | Julia Kozlitina<br>Helen H Hobbs |

The funders had no role in study design, data collection and interpretation, or the decision to submit the work for publication.

### Author contributions

RSC, SAI, Conception and design, acquisition of data, analysis and interpretation of data, drafting or revising the article; JK, Analysis and interpretation of data, drafting or revising the article; AG,

Conception and design, design and acquisition of data, analysis and interpretation of data, drafting or revising the article; JCC, AR, HHH, Conception and design, analysis and interpretation of data, drafting or revising the article

## Author ORCIDs

Arun Radhakrishnan, http://orcid.org/0000-0002-7266-7336
Helen H Hobbs, http://orcid.org/0000-0002-8700-9897

## Ethics

Human subjects: All study protocols were approved by the Institutional Review Board (IRB) of the University of Texas Southwestern Medical Center, and all subjects provided written informed consent. Each participant completed a detailed staff-administered survey, including questions about socioeconomic status, medical history and medication use. Ancestry was self-reported.

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
