## [Decision Letter]

Thank you for submitting your work entitled "Variable accessibility of cholesterol in red blood cell membranes among humans" for consideration by *eLife*. Your article has been reviewed by three peer reviewers, one of whom is a member of our Board of Reviewing Editors, and the evaluation has been overseen by a Senior Editor. The reviewers have opted to remain anonymous.

Our decision has been reached after consultation between the reviewers. Based on these discussions and the individual reviews below, we regret to inform you that your manuscript will not be considered further for publication in *eLife* in its present form.

All three reviewers agreed the current version of the manuscript was not acceptable for *eLife*. However, each reviewer thought that the general topic was novel and potentially important. The main concern is that the observations were predictable and/or descriptive. Also, the link between the observations and plasma lipid metabolism was not strong. Each of the three reviewers had suggestions for improvements (see comments below). I believe that all of the reviewers would be interested in reviewing a revised version of the manuscript at a later time point, but only if the concerns were fully addressed.

Reviewer #1:

The group at UT-Southwestern has already published groundbreaking biochemistry on the binding of cytolysins to cholesterol in the plasma membrane and endoplasmic reticulum. Some of the experiments in the current study in erythrocytes shows the validity of earlier biochemical findings (for example, reduced binding with cyclodextrin and more binding after sphingomyelinase treatment). What is nice (and novel) is extending studies with ALOD4 to a mammalian cell type (red blood cells). A lot of the paper is descriptive, showing that the binding depends on ethnic background, the presence of renal disease, HDL levels, triglyceride levels, and phospholipid species in the erythrocyte membrane. However, despite this important progress, a lot of the variation in binding of ALOD4 remains unexplained. The authors speculated that the accessibility of cholesterol to ALOD4 is relevant to plasma cholesterol transport, but a strong connection was not made.

A plus is the novelty of this line of investigation, and the promise that it will be the beginning of a new chapter in understanding plasma membrane cholesterol and plasma cholesterol transport. However, there are some negatives. One is that a lot of the results are predictable from the earlier papers from UT Southwestern. Another is that the work, although quite interesting, remains largely descriptive. A stronger link to metabolism is needed. Also, I would like to see quantitative analysis of ALOD4 binding to individual erythrocytes. They show median levels of binding. What is the variation in binding to different erythrocytes? What is the level of heterogeneity in ALO binding to erythrocytes? What is the correlation between RFP-ALO binding and GFP-ALO binding in the same experiment? What accounts for variation in binding? If some erythrocytes in a population display more ALO binding than others, then I would like to see some explanation for the difference. Do cells that bind more ALO have more plasma membrane cholesterol? Very high resolution images of the binding of fluorescent ALO are needed.

Reviewer #2:

This paper presents the very interesting finding that binding of a fluorescent anthrolysin O derivative to red blood cells varies by about 10-fold in individuals – even in individuals without known health issues. This fALOD4 binding is reflective of "accessible" cholesterol in the exofacial leaflet of the RBCs. It is not correlated with total cholesterol, which indicates that other features of the RBC membrane are responsible for these differences. This suggests the possibility that differences in RBC accessible cholesterol could relate to other parameters such as reverse cholesterol transport.

While this is a very interesting and potentially important observation, the rest of the paper does little to explain mechanistically how this difference is driven or what the consequences of high or low accessible cholesterol might be.

There were some general issues and some technical issues with the paper, which are listed below separately.

Overall. I found the paper reports one very interesting observation but it does not provide either a clear mechanism for the phenomenon or a clear relation to human health.

Technical

1) The dependence on number of cells in the assay (Figure 3) is very peculiar and needs to be explained. Individual 1 and 2 actually cross at high RBC concentrations. Is it possible that the fALDO4 is being depleted? Why does the concentration dependence vary so dramatically for different people? This phenomenon, which is not explained, could lead to variations between samples.

2) It is very disappointing that the fatty acid saturation of the lipids was not reported in the lipidomics analysis.

3) It is surprising that PA has such a large effect since it is reported to be a very minor lipid (0.4%). Also SM is described as affecting fALOD4 binding even though the text states it has 0.01% effect. These analyses along with the lack of measurement of lipid saturation leave one with very little understanding of what is responsible for difference in accessible cholesterol.

General

1) In the studies comparing ethnic groups, there were large disparities in% male and in the age distribution among the groups. While it is claimed that there were adjustments made to account for these differences, it is not at all clear that there are enough data to make such adjustments with high confidence.

2) The authors state that the relationship between blood type and fALOD4 binding will require further sampling. Since no firm conclusions were drawn aside from an unexplained Rh correlation, it isn't clear why this is included.

3) The authors state that the populations used in Table 2 and Table 5 show a similar trend in fALOD4 binding. It would be helpful to have a more complete analysis of the similarities and differences in these populations.

4) Other studies have found that differences in diet can affect RBC lipid composition (e.g., DOI: 10.5584/jiomics.v3i1.123). This may have an unknown impact on the differences among ethnic groups, and it could lead to non-genomic influences on the accessible cholesterol.

Reviewer #3:

This manuscript by Hobbs and colleagues describes methods for measuring the chemical activity of cholesterol in human RBCs. Using this assay, the authors report several interesting, new observations. Notably, the chemical activity of cholesterol in RBCs: (1) is stable within fasted individuals although there is variance among cells, (2) varies 10-fold between individuals, (3) positively correlates with HDL-C and Rh antigen, (4) negatively correlates with plasma TG, (5) is higher in patients on hemodialysis but unaffected acutely by the procedure, and (6) that Tangiers individuals have extremely low chemical activity. In addition, they report that RBCs behave like other cells in having a threshold for fugacity and that depletion of phospholipids and SM increase the chemical activity of cholesterol. The manuscript is well-written, experiments employ appropriate statistical tests, the methods are clear, and the conclusions are supported by the data.

However, the general significance and impact of this study is moderate. The authors discuss that RBCs may participate in RCT and therefore this assay may be a useful tool in these studies, but experiments in the manuscript do not address this exciting hypothesis. The assay is robust, but no data indicate its predictive value. Furthermore, no experiments test whether chemical activity changes within an individual which would suggest that it may play a role in pathological processes. For example, does chemical activity change in response to refeeding or diet? I am left asking, what open question was answered by these studies?

In summary, this is an excellent study that describes the development of an assay that in the future will be used to test an important hypotheses

---

## [Author Response]

All three reviewers agreed the current version of the manuscript was not acceptable for eLife. However, each reviewer thought that the general topic was novel and potentially important. The main concern is that the observations were predictable and/or descriptive. Also, the link between the observations and plasma lipid metabolism was not strong. Each of the three reviewers had suggestions for improvements (see comments below). I believe that all of the reviewers would be interested in reviewing a revised version of the manuscript at a later time point, but only if the concerns were fully addressed.

The main concern of the reviewers was that “the observations were predictable and/or descriptive. Also, the link between the observations and plasma lipid metabolism was not strong.”

We concur in part with these comments of the reviewers. The basic characteristics of cholesterol accessibility in RBC membranes, as determined by the fALOD4 assay, are consistent with those described in the literature. We included these experiments to show that the assay we have developed to analyze RBCs from humans has the same characteristics as similar assays that have been used to analyze other membranes and cellular organelles. The fact that the behavior of the assay is consistent with predictions supports the contention that we are measuring cholesterol accessibility.

However, it is not obvious to us that the major findings reported in this paper are predictable. To our knowledge this is the first time that the cholesterol accessibility in RBC membranes has been evaluated as a phenotype in humans, and the finding that fALOD4 binding differs over a fold range among healthy individuals was not predicted previously. This amount of variability exceeds that seen for most quantitative traits in humans (for example, levels of plasma HDL-C or levels of circulating liver enzymes).

We agree with the reviewers, however, that the results described in this paper are largely descriptive. Like many quantitative traits that vary between individuals, many factors, both genetic and nongenetic, contribute to the variability. In this paper we have identified some of the factors contributing either directly or indirectly to the variability in RBC cholesterol accessibility in humans. Importantly, we show that the cholesterol accessibility is not correlated with RBC cholesterol content. However, due to the largely descriptive nature of the paper, we ask that the paper now be considered for publication in the Tools and Resources category.

We have changed the title accordingly to:

“Variability of cholesterol accessibility in human red blood cells measured using a bacterial cholesterol-binding toxin”

We deleted the data from the Tangier and the LCAT deficient families since it is not pertinent to the description of the assay and the characterization of factors contributing to variation in the assay and in the trait in humans.

Reviewer #1:

The group at UT-Southwestern has already published groundbreaking biochemistry on the binding of cytolysins to cholesterol in the plasma membrane and endoplasmic reticulum. Some of the experiments in the current study in erythrocytes shows the validity of earlier biochemical findings (for example, reduced binding with cyclodextrin and more binding after sphingomyelinase treatment). What is nice (and novel) is extending studies with ALOD4 to a mammalian cell type (red blood cells).

We agree with the reviewer.

A lot of the paper is descriptive, showing that the binding depends on ethnic background, the presence of renal disease, HDL levels, triglyceride levels, and phospholipid species in the erythrocyte membrane. However, despite this important progress, a lot of the variation in binding of ALOD4 remains unexplained. The authors speculated that the accessibility of cholesterol to ALOD4 is relevant to plasma cholesterol transport, but a strong connection was not made.

We also agree with the statement that the paper is descriptive (see above), but we think that the assay is sufficiently robust and the results sufficiently provocative that publication will stimulate further studies using this system.

Our speculations are confined to the Discussion.

A plus is the novelty of this line of investigation, and the promise that it will be the beginning of a new chapter in understanding plasma membrane cholesterol and plasma cholesterol transport.

This was the intent of our paper. The paper provides the tools and rationale to address these questions.

However, there are some negatives. One is that a lot of the results are predictable from the earlier papers from UT Southwestern.

Please see our response to the general comments. The major findings of the paper are not predictable. To our knowledge, it had not been anticipated that the accessibility in RBC cholesterol levels would vary so widely among humans, and that this variation was not due to a variation in cholesterol concentration. We think the only predictable results in the paper are in Figure 1, which is required to validate the assay, and Figure 7, which was performed as a complement to our analysis of the relationship between phospholipid composition and fALOD4 binding to RBCs.

*Another is that the work, although quite interesting, remains largely descriptive. A stronger link to metabolism is needed.*

As discussed above, we think that this is a fair comment and have responded by changing the classification of the paper to a Tool and Resource.

Also, I would like to see quantitative analysis of ALOD4 binding to individual erythrocytes.

We utilize a FACS to perform the analysis. We have now provided the distribution of fALOD4 binding in representative individuals with low, medium and high binding (Figure 3). We cannot provide the absolute values for single RBCs since the FACS provides the information in Relative Units.

*They show median levels of binding. What is the variation in binding to different erythrocytes? What is the level of heterogeneity in ALO binding to erythrocytes?*

We have added examples of the distribution in fALOD4 binding for representative individuals (Figure 3). Representative data from another individual is also shown in Figure 1.

What is the correlation between RFP-ALO binding and GFP-ALO binding in the same experiment?

We have not performed experiments with either RFP-ALO or GFP- ALO. Initially when we set up the assay we also conjugated ALOD4 to Alexa 647, a far red fluorescent dye, and the results of the assay were very similar. Neither Alexa 488 or Alexa 647 alone confers any fluorescence to RBCs.

What accounts for variation in binding? If some erythrocytes in a population display more ALO binding than others, then I would like to see some explanation for the difference. Do cells that bind more ALO have more plasma membrane cholesterol?

We show the median level of fALOD4 binding is not related to the mean absolute level of membrane cholesterol in Figure 3. We also have shown that the median levels of accessible cholesterol are not related to the morphology of the RBC or to the pronase-sensitive proteins on the surface of erythrocytes.

Very high resolution images of the binding of fluorescent ALO are needed.

We agree that such images may be interesting and informative but they are beyond the scope of this paper. We think it unlikely that such images would provide significant mechanistic insights without extensive additional studies.

Reviewer #2:

This paper presents the very interesting finding that binding of a fluorescent anthrolysin O derivative to red blood cells varies by about 10-fold in individuals – even in individuals without known health issues. This fALOD4 binding is reflective of "accessible" cholesterol in the exofacial leaflet of the RBCs. It is not correlated with total cholesterol, which indicates that other features of the RBC membrane are responsible for these differences. This suggests the possibility that differences in RBC accessible cholesterol could relate to other parameters such as reverse cholesterol transport.

While this is a very interesting and potentially important observation, the rest of the paper does little to explain mechanistically how this difference is driven or what the consequences of high or low accessible cholesterol might be.

We agree that we have not fully explained the mechanistic basis of the variation in RBC cholesterol accessibility. We are not aware of a human quantitative trait for which the variation in levels has been fully explained. As a comparison, plasma levels of LDL-C vary over a 4-5 fold range in the general population and despite this trait first being shown to vary in humans over 65 years ago, we still cannot account for most of the factors contributing to its variability. Here we describe a new trait that varies over 10-fold and have identified factors that explain ~50% of the variability.

Our intent in this paper is to describe a well-validated assay that provides a new tool to examine an understudied aspect of sterol metabolism and trafficking. A major strength of the assay is that it can be performed easily using a readily accessible tissue in large numbers of humans.

We have speculated in the discussion how differences in RBC cholesterol accessibility may be related to both cholesterol trafficking and to cholesterol homeostasis (and disease) in humans. Testing these hypotheses will be the focus of subsequent papers from us, and hopefully from others.

There were some general issues and some technical issues with the paper, which are listed below separately.

Overall. I found the paper reports one very interesting observation but it does not provide either a clear mechanism for the phenomenon or a clear relation to human health.

We essentially agree with these comments. We have made no claims in the paper regarding any relationship to human health.

Technical

1) The dependence on number of cells in the assay (Figure 3) is very peculiar and needs to be explained. Individual 1 and 2 actually cross at high RBC concentrations. Is it possible that the fALDO4 is being depleted? Why does the concentration dependence vary so dramatically for different people? This phenomenon, which is not explained, could lead to variations between samples.

The reviewer raises an excellent point. Depletion of the fALOD4 probe could lead to variations in fALOD4 binding. Figure 3 addresses this possibility, and we have rewritten the last paragraph of subsection “RBC cholesterol accessibility varied between individuals in a reproducible and stable manner” to clarify our interpretation of the figure. We have also flipped the x-axis to make the figure more intuitive (going from fewer to more RBCs. A fixed amount of fALOD4 was used in the assay, and an ~5-fold variation in fALOD4 binding/cell was observed between individuals 1, 2, and 3. This variation is only observed when the RBCs/reaction ranged from 105 to 106.

When more RBCs were added in the reaction, fALOD4 binding/cell decreased in all 3 cases, likely because the fALOD4 probe became limiting. In the rest of the studies, we avoided these extreme ratios of RBCs to fALOD4 where the probe could be limiting, and used 2.5 x 105 RBCs/assay.

2) It is very disappointing that the fatty acid saturation of the lipids was not reported in the lipidomics analysis.

We have now added this data as a supplementary table, which was not very revealing, but given the large number of fatty acids, some of which are present in very low concentrations, additional studies with a larger number of individuals will need to be performed to rule out any effect of fatty acid levels on fALOD4 binding.

3) It is surprising that PA has such a large effect since it is reported to be a very minor lipid (0.4%). Also SM is described as affecting fALOD4 binding even though the text states it has 0.01% effect. These analyses along with the lack of measurement of lipid saturation leave one with very little understanding of what is responsible for difference in accessible cholesterol.

We tested the most likely parameters to influence cholesterol accessibility in RBC membranes, which include the intrinsic properties of the RBCs, the plasma lipoprotein levels, as well as protein and phospholipid composition of the RBC membrane. Since this analysis was based on statistical regression procedures, it is by necessity exploratory and correlative. At present, we do not know if the correlation with PA in vivo is primary or secondary. PA levels may be correlated with a more abundant lipid or combination of lipids that determine accessible cholesterol content. Due to extensive inter-correlations among the different phospholipid levels, it is not possible to disentangle the causal and correlative effects of these lipids using statistical tools alone, and other experiments would be needed to determine the true magnitude of these effects.

General

1) In the studies comparing ethnic groups, there were large disparities in% male and in the age distribution among the groups. While it is claimed that there were adjustments made to account for these differences, it is not at all clear that there are enough data to make such adjustments with high confidence.

Although we adjusted for age and gender in our analyses, neither factor was a major determinant of fALOD4 binding (Table 2 and Table 5). Nonetheless, to ensure that disparities in these factors did not bias our main findings, we have performed sensitivity analyses stratified by ethnicity. Similar to our main analysis, age and gender were not major predictors of fALOD4 binding within ethnic groups, as shown in Table 7. Age was not significantly associated with fALOD4 binding, and male gender was weakly associated with higher fALOD4 binding, but only in Whites (p=0.038). Thus, it is unlikely that our main conclusions are confounded by differences in age and gender between ethnic groups.

Similarly, we have repeated the correlation analysis with RBC lipids stratified by ethnicity. In this analysis, LPE and SM were the only lipid classes consistently and significantly associated with fALOD4 binding in both Blacks and Hispanics, after adjustment for demographic factors, and the association for PA approached significance in both groups (Table 8). Therefore, it is perhaps not surprising that LPE, SM, and PA, remained significant predictors of fALOD4 binding in multivariate analysis.

Author response table 1.Factors associated with fALOD4 binding in 364 healthy subjects, stratified by ethnicity.**DOI:**
http://dx.doi.org/10.7554/eLife.23355.019**Ethnicity****Factor****Β****SE****P-value**White (n=182)Age0.0410.0300.1766Male gender0.0970.0650.1351Black (n=98)Age-0.0980.0570.0863Male gender0.0200.1040.8503Hispanic (n=84)Age0.0310.0430.4716Male gender0.0370.0910.6886

Author response table 2.Correlation of RBC lipids with fALOD4 binding in 123 subjects in whom RBC membrane lipids were measured.**DOI:**
http://dx.doi.org/10.7554/eLife.23355.020LipidAA (n=41)HA (n=82)classPartial rP-valuePartial rP-valueCer0.280.07620.020.8615ePC-0.150.3414-0.030.7717ePE-0.170.28170.170.1331ePS0.240.13510.130.2465HexCer-0.140.3796-0.090.4333**LPC****0.35****0.0250**0.160.1613**LPE****-0.46****0.0023****-0.32****0.0029**PA**0.33****0.0361**0.210.0599PC-0.130.4199-0.190.0882PE-0.180.2542-0.150.1726**PG****-0.41****0.0078**0.070.5341**PI****-0.35****0.0264**-0.070.5052PS-0.170.2983-0.120.2704**SM****0.40****0.0094****0.33****0.0024**Partial correlation coefficients (r) were calculated using linear regression adjusted for age and gender. Bold font indicates correlations significant at p<0.05.

We have now repeated all of our analyses adjusting for age, sex and race. The only significant change we found was that the relationship between fALOD4 binding and plasma HDL-C levels did not reach statistical significance (see revised Figure 5).

2) The authors state that the relationship between blood type and fALOD4 binding will require further sampling. Since no firm conclusions were drawn aside from an unexplained Rh correlation, it isn't clear why this is included.

The data is included because it complements our analysis of the effect of pronase treatment of RBCs on fALOD4 binding. Rh antigen provides an example of a membrane protein that appears to correlate with fALOD4 binding. Others have shown that Rh antigen is resistant to pronase (Satchell TJ et al., Blood 118:182, 2011). There may be other such pronase resistant proteins that contribute to inter-individual differences in RBC cholesterol accessibility. As reviewed in the Discussion, performing an unbiased genetic screen such as a genome wide association study in a population study may reveal other factors that contribute to the variability in the trait.

3) The authors state that the populations used in Table 2 and Table 5 show a similar trend in fALOD4 binding. It would be helpful to have a more complete analysis of the similarities and differences in these populations.

We opted to keep the analysis of the two cohorts separate to avoid technical artifacts due to differences in the time of measurement and other uncontrolled factors. However, we observed very similar ethnic differences in fALOD4 binding between the two populations. In both cohorts, Blacks had significantly higher fALOD4 binding than either Whites or Hispanics (Whites were not included in the manuscript, because phospholipid measurements were not available for this sub-group). Age and gender were not major predictors of fALOD4 binding, and triglycerides were significantly inversely correlated with fALOD4 binding. We believe that the replication of the results across two independent cohorts further increases the confidence in our findings.

4) Other studies have found that differences in diet can affect RBC lipid composition (e.g., DOI: 10.5584/jiomics.v3i1.123). This may have an unknown impact on the differences among ethnic groups, and it could lead to non-genomic influences on the accessible cholesterol.

We have now addressed this possibility and added the following sentence to the Discussion. “Dietary differences between the ethnic groups may contribute to differences in RBC cholesterol accessibility.”

Reviewer #3:

This manuscript by Hobbs and colleagues describes methods for measuring the chemical activity of cholesterol in human RBCs. Using this assay, the authors report several interesting, new observations. Notably, the chemical activity of cholesterol in RBCs: (1) is stable within fasted individuals although there is variance among cells, (2) varies 10-fold between individuals, (3) positively correlates with HDL-C and Rh antigen, (4) negatively correlates with plasma TG, (5) is higher in patients on hemodialysis but unaffected acutely by the procedure, and (6) that Tangiers individuals have extremely low chemical activity. In addition, they report that RBCs behave like other cells in having a threshold for fugacity and that depletion of phospholipids and SM increase the chemical activity of cholesterol. The manuscript is well-written, experiments employ appropriate statistical tests, the methods are clear, and the conclusions are supported by the data.

However, the general significance and impact of this study is moderate. The authors discuss that RBCs may participate in RCT and therefore this assay may be a useful tool in these studies, but experiments in the manuscript do not address this exciting hypothesis.

We agree with this reviewer. Our discussion regarding the possible role of RBC in reverse cholesterol transport builds upon earlier studies that have suggested this possibility. We have limited our speculation to the Discussion.

The assay is robust, but no data indicate its predictive value.

We agree that the predictive value of the assay remains to be demonstrated. The most exciting possibility is that variation in cholesterol accessibility will help to explain why some conditions, such as chronic renal failure and diabetes, are associated with an elevated risk of coronary heart disease that cannot be explained by established risk factors. For this reason we elected to study fALOD4 binding in RBCs from individuals with chronic renal failure. We recognize that our results, though encouraging, are by no means definitive, and we present them as such.

Furthermore, no experiments test whether chemical activity changes within an individual which would suggest that it may play a role in pathological processes. For example, does chemical activity change in response to refeeding or diet? I am left asking, what open question was answered by these studies?

We agree that such studies would be potentially interesting, but are beyond the scope of this paper. Given that our fatty acid analysis revealed no correlation with chemical activity, we think it is unlikely that the differences in dietary intake contribute substantially to the differences observed.

In summary, this is an excellent study that describes the development of an assay that in the future will be used to test an important hypotheses

We accept this assessment and ask that the paper be considered for publication in the Tools and Resources category.